# The threefold way to quantum periods: WKB, TBA equations and q-Painlevé

**Fabrizio Del Monte[1,2]⋆ and Pietro Longhi[3,4]†**

**1** Centre de Recherches Mathématiques, Université de Montréal,
C. P. 6128, Succ. Centre Ville, Montréal, QC H3C 3J7 Canada
**2** Department of Mathematics and Statistics, Concordia University,
1455 de Maisonneuve Blvd. W. Montréal, QC H3G 1M8 Canada
**3** Department of Physics and Astronomy, Uppsala University,
Box 516, 751 20 Uppsala, Sweden
**4** Department of Mathematics, Uppsala University, Box 480, 751 06 Uppsala, Sweden

⋆ delmonte@crm.umontreal.ca , † pietro.longhi@physics.uu.se

## Abstract

We show that TBA equations defined by the BPS spectrum of 5d $\mathcal{N} = 1$ $SU(2)$ Yang-Mills on $S^1 \times \mathbb{R}^4$ encode the q-Painlevé III$_3$ equation. We find a fine-tuned stratum in the physical moduli space of the theory where solutions to TBA equations can be obtained exactly, and verify that they agree with the algebraic solutions to q-Painlevé. Switching from the physical moduli space to that of stability conditions, we identify two one-parameter deformations of the fine-tuned stratum, where the general solution of the q-Painlevé equation in terms of dual instanton partition functions continues to provide explicit TBA solutions. Motivated by these observations, we propose a further extensions of the range of validity of this correspondence, under a suitable identification of moduli. As further checks of our proposal, we study the behavior of exact WKB quantum periods for the quantum curve of local $\mathbb{P}^1 \times \mathbb{P}^1$.



# 1 Introduction

This paper puts forward a new approach to the exact solution of TBA equations of the type studied in [1], applied to BPS structures of *five-dimensional* supersymmetric QFTs. We obtain exact solutions by mapping the problem of solving the complicated TBA integral equations to an appropriate q-Painlevé equation [2], whose general solution is known to be written in terms of five-dimensional Nekrasov functions [3–9]. In doing so, we also propose an explicit connection to the monodromy theory of difference equations arising from quantum mirror curves in Topological Strings.

**The general picture**

Our starting point is the five-dimensional Seiberg-Witten description [10, 11], where the Seiberg-Witten curve $\Sigma$ is identified with the mirror curve of the local Calabi-Yau geometry "geometrically engineering" the 5d theory on $\mathbb{R}^4 \times S^1$ [12, 13]. In this geometric description, a stable BPS state is a calibrated cycle $\gamma \in H_1(\Sigma)$, and its central charge is computed by the period of the Seiberg-Witten differential $\lambda_{SW}$ along the cycle $\gamma$. The low energy description has further instantonic corrections once the theory is compactified on a second circle. These corrections are encoded by a system of TBA equations derived in physics [1,14–16], and reformu-

lated mathematically by Bridgeland in terms of a Riemann-Hilbert Problem (RHP) associated to BPS structures [17, 18]:

$$\log Y_\gamma(\epsilon) = \frac{Z_\gamma}{\epsilon} - \frac{\epsilon}{\pi i} \sum_{\gamma' > 0} \Omega(\gamma', u)\langle\gamma, \gamma'\rangle \int_{\ell_{\gamma'}} \frac{d\epsilon'}{(\epsilon')^2 - (\epsilon)^2} \log(1 - \sigma(\gamma')Y_{\gamma'}(\epsilon')). \tag{1}$$

Equation (1) can be iteratively solved, yielding an asymptotic series in $\epsilon$. Viewing this as a quantum deformation of the periods of $\lambda_{SW}$, we refer to $\log Y_\gamma$ loosely as "quantum periods". In the case of four-dimensional theories, the connection between TBAs and WKB quantum periods is very well studied, and it leads to the statement that $Y_\gamma$ indeed coincide with monodromy coordinates of differential equations on Riemann Surfaces (opers) [19]. This correspondence has motivated many recent studies of $Y_\gamma$ by means of exact WKB methods [20, 21]. The differential equations are obtained by quantizing the corresponding Seiberg-Witten curve, and are often called quantum curves.

In the five-dimensional case relevant to this paper, the Seiberg-Witten/mirror curve is defined over $\mathbb{C}^* \times \mathbb{C}^*$, and as a result the quantum mirror curve is a difference, rather than a differential, equation. Indeed, the curve and differential arise as the leading order WKB approximation of a difference equation, whose solution is the open (refined) topological string partition function in the Nekrasov-Shatashvili limit [22, 23]. This leads us across the second road on our journey, namely WKB approximation of difference equations. As in four dimensions, the general expectation is that the $\epsilon$-expansion of (1) and the WKB expansion of the quantum periods coincides, after an appropriate matching of parameters.

TBA and WKB are by now relatively traditional approaches to the study of quantum periods, dating back to the seminal works of Gaiotto Moore and Neitzke [1, 24].[1] Unfortunately, each of these frameworks becomes computationally prohibitive when applied to five-dimensional theories. This is especially true of those theories engineered by Calabi-Yau threefolds with compact divisors, due to wall-crossing phenomena in the BPS spectrum (see [16, 27–29] for previous results on the case without compact divisors).

In this paper we chart a third route, mapping the problem to a q-Painlevé equation that allows us to obtain explicit solutions even in theories characterized by the richness of 5$d$ wall-crossing phenomena. The three approaches we just described are *a priori* different quantizations of the classical periods describing BPS states of the five-dimensional QFT, and we provide evidence for their identification with a precise map of the parameters involved. The relation between them is outlined in Figure (1).

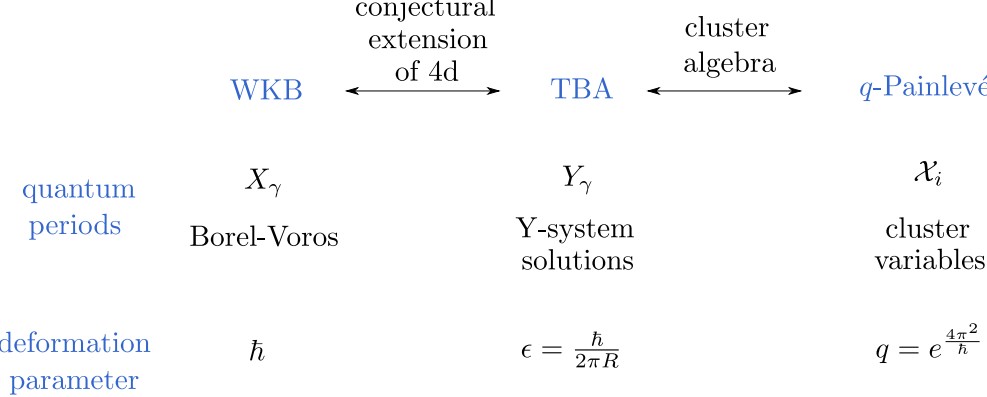

Figure 1: The general picture.

---

**Main results**

For illustration purposes, we focus on M-theory in the Calabi-Yau background of local $\mathbb{P}^1 \times \mathbb{P}^1$, which engineers 5d $\mathcal{N} = 1$ Super Yang-Mills theory with gauge group $SU(2)$ [30–33]. The four-dimensional limit of this theory, 4d $\mathcal{N} = 2$ $SU(2)$ Yang-Mills, has a Seiberg-Witten description [10] based on the spectral curve of the Toda chain [34, 35]. The 5d theory also admits a Seiberg-Witten description [11], corresponding to a relativistic deformation of the Toda chain [36], with curve

$$F(e^x, e^y) = \tau(e^x + e^{-x}) + e^y + e^{-y} - \kappa = 0. \tag{2}$$

The BPS charge lattice in this case is four-dimensional, and we denote its generators by $\gamma_1, \ldots, \gamma_4$. In this paper we show that the moduli space of quantum mirror curves of local $\mathbb{P}^1 \times \mathbb{P}^1$ contains a locus, which we call the *fine-tuned stratum* $\mathcal{C}_1^{(0)}$, which is characterized by a $\mathbb{Z}_2$ symmetry acting on the classical (and quantum) periods:

$$\mathcal{C}_1^{(0)}: \quad Z_{\gamma_1} = Z_{\gamma_3}, \quad Z_{\gamma_2} = Z_{\gamma_4}, \quad \arg Z_{\gamma_1} > \arg Z_{\gamma_2}, \quad Z_{\gamma_1 + \gamma_2} \in \mathbb{R}^+. \tag{3}$$

The locus belongs to a larger BPS chamber, where the spectrum of BPS states is remarkably simple and characterized by an affine symmetry [37–39]. On the one hand, BPS states govern both discontinuities of TBA solutions $Y_\gamma$, and the Stokes automorphisms of Borel-resummed quantum WKB periods $X_\gamma$. On the other hand, the BPS spectrum is encoded in the discrete-time evolution of a cluster integrable system described by the q-Painlevé $III_3$ equation, whose solutions will be denoted $\mathcal{X}_i$. This implies that, up to the identification of suitable boundary conditions for q-Painlevé, the TBA solutions / WKB quantum periods must satisfy the q-Painlevé equations. This perspective is fruitful for the geometry we study, since a solution of q-Painlevé equations was obtained in [3–5, 8] in terms of 5d instanton partition functions. By matching moduli of the 5d gauge theory with those of the quantum mirror curve, we identify the discrete time evolution of q-Painlevé with a trajectory in the parameter space of the latter, leading to the identification of boundary conditions for the equation from the degeneration of the curve into two 'half-geometries' (see Figure 6).

After establishing a dictionary between the q-Painlevé $III_3$ equation and the quantum mirror curve moduli, we translate solutions of the former into quantum periods for the latter. Remarkably we find that on the fine-tuned stratum (3) the quantum periods can be computed exactly: due to the $\mathbb{Z}_2$ symmetry, all $\epsilon$-corrections in the TBA equations (1) cancel out and the semiclassical answer is exact, yielding

$$\mathcal{C}_1^{(0)}: \qquad Y_{\gamma_1} = Y_{\gamma_3} = e^{\frac{\pi}{R\epsilon}} \tau^{-\frac{i}{R\epsilon}}, \qquad Y_{\gamma_2} = Y_{\gamma_4} = \tau^{\frac{i}{R\epsilon}}. \tag{4}$$

Here $\tau$ is a modulus of the mirror (Seiberg-Witten) curve (2) and $R$ is the radius of the M-theory circle. We check that the same holds for the solutions to q-Painlevé in a suitable limit of its moduli, finding exact agreement with its class of *algebraic solutions* [3, 4]. There is a very close analogy between the definition of the fine-tuned stratum and that of algebraic solution, so it is natural to expect that this observation holds for more general cases: we formulate this as Conjecture 1.

We also study two deformations of the fine-tuned stratum in the space of stability conditions (see Section 2.2.3), that we denote $\mathcal{C}_1^{(\delta)}$ and $\mathcal{C}_1^{(\rho)}$ respectively. While $\mathcal{C}_1^{(0)}$ belongs to the physical moduli space of the theory (parameterized by the classical curve parameters $\tau, \kappa$), it isn't clear at the moment if this is true also for $\mathcal{C}_1^{(\delta)}, \mathcal{C}_1^{(\rho)}$, or if they belong only to the moduli space of stability conditions. Nevertheless we argue that, while either deformation breaks the $\mathbb{Z}_2$, it does not induce wall-crossing of the BPS spectrum. This means that $\mathcal{C}_1^{(\delta)}, \mathcal{C}_1^{(\rho)}$ belong to

the same BPS chamber as the fine-tuned stratum, that we call the collimation chamber $\mathcal{C}_1$, following our earlier work [39]. Differently from what happened in the fine-tuned case, quantum corrections in the TBA equations (1) no longer cancel out. The solutions are now

$$
Y_{\gamma_1} = (qt)^{1/2} \left( \frac{Z_D(u,s,q,qt)}{s^{\frac{1}{2}} Z_D(q^{\frac{1}{2}}u,s,q,qt)} \right)^2, \quad Y_{\gamma_2} = t^{-\frac{1}{2}} \left( \frac{s^{\frac{1}{2}} Z_D(q^{\frac{1}{2}}u,s,q,t)}{Z_D(u,s,q,t)} \right)^2,
$$

$$
Y_{\gamma_3} = (qt)^{1/2} \left( \frac{s^{\frac{1}{2}} Z_D(q^{\frac{1}{2}}u,s,q,qt)}{Z_D(u,s,q,qt)} \right)^2, \quad Y_{\gamma_4} = t^{-\frac{1}{2}} \left( \frac{Z_D(u,s,q,t)}{s^{\frac{1}{2}} Z_D(q^{\frac{1}{2}}u,s,q,t)} \right)^2,
\tag{5}
$$

where $Z_D(u,s,q,t)$ is the dual instanton partition function of the 5d theory (see Section 5), $t := Y_{\gamma_2}^{-1} Y_{\gamma_4}^{-1}$, $q := Y_{\gamma_1} Y_{\gamma_2} Y_{\gamma_3} Y_{\gamma_4}$, and $u,s$ take the following values:

$$
\mathcal{C}_1^{(\delta)} : \begin{cases} u^2 = e^{\frac{2\pi^2}{\hbar}(1+O(\hbar))}, \\ s = e^{-\frac{\pi R}{\hbar}\delta(1+O(\hbar))} \times (\text{nonpert. corrections in } \epsilon), \end{cases}
\tag{6}
$$

$$
\mathcal{C}_1^{(\rho)} : \begin{cases} u^2 = e^{\frac{4\pi^2}{\hbar(1+\rho)}(1+O(\hbar))}, \\ s = e^{O(\hbar^0)} \times (\text{nonpert. corrections in } \hbar). \end{cases}
\tag{7}
$$

When $\delta = \rho = 0$, the factors involving $Z_D$ in (5) simplify, and one is left with the algebraic solution (4) after appropriate matching of $q, t$ with $\tau, \hbar$.

The proposed identification of solutions to TBA equations and solutions of q-Painlevé requires some care. For instance, note that Nekrasov functions in (5) are single-valued functions of $q = e^{\frac{4\pi^2}{\hbar}}$ while the solutions to TBA equations have jumps with variations of $\arg \hbar$. This is explained in Remark 3 by interpreting (5) as a particular resummation of the perturbative expansion in $\hbar$.

The paper is organized as follows. Section 2 collects some background on the geometry, and known results about the BPS spectrum. Here we also include a novel observation concerning the existence of the fine-tuned stratum (hence of the collimation chamber) in the physical moduli space. In Section 3 we discuss the computation of quantum periods via exact WKB analysis for difference equations. In Section 4 we formulate the TBA equations in the conformal limit, for the BPS spectrum corresponding to the collimation chamber. We give the exact solution on the fine-tuned stratum, where the equations essentially decouple. In Section 5 we recall the connection between 5d gauge theory, BPS states and q-Painlevé equations, which leads us to a new characterization of quantum periods in terms of 5d instanton partition functions. Section 6 collects concluding remarks and open directions. Appendices contain additional material: an exponential network analysis of the fine-tuned stratum, an analysis of the half-geometry limit, and computations of WKB quantum periods for first-order $\hbar$-difference equations.

## 2 Classical geometry and classical periods

### 2.1 Local $\mathbb{P}^1 \times \mathbb{P}^1$ and its mirror

Five-dimensional SCFTs of rank one arise via geometric engineering in M-theory on local del Pezzo and Hirzebruch surfaces [12, 30–33]. For illustration purposes we will focus on the $E_1$ model, corresponding to the fixed point of 5d $\mathcal{N} = 1$ $SU(2)$ Yang-Mills. This theory is engineered by considering M-theory in the background of the local Hirzebruch surface $\mathbb{P}^1 \times \mathbb{P}^1$. The complexified Kähler moduli, corresponding to the areas of the two $\mathbb{P}^1$'s, are related to the

Coulomb modulus and the dimensionful gauge coupling of the theory. An additional modulus arises by formulating the theory on $S^1 \times \mathbb{R}^4$, and corresponds to the radius of the circle.

A five-dimensional QFT on $S^1$ may be viewed as a four-dimensional theory of its Fourier modes, also known as a Kaluza-Klein (KK) 4d $\mathcal{N} = 2$ theory [37, 40]. The mirror Calabi-Yau $X^\vee$ is the hypersurface

$$uv = F(e^x, e^y), \tag{8}$$

describing a bundle of conics over $\mathbb{C}^* \times \mathbb{C}^*$, with fiber that degenerates over the mirror curve $\Sigma$ described by

$$F(e^x, e^y) = \tau(e^x + e^{-x}) + e^y + e^{-y} - \kappa = 0. \tag{9}$$

The mirror curve is topologically a torus with four punctures, and coincides with the Seiberg-Witten curve for the $E_1$ theory [11]. As usual in Seiberg-Witten descriptions, certain one-cycles on $\Sigma$ correspond to charges of BPS states and the central charge is computed by periods of a one-form $\lambda$

$$Z_\gamma = \oint_\gamma \lambda, \tag{10}$$

with

$$\lambda = \frac{1}{2\pi R}\, y\, dx. \tag{11}$$

In general, the periods can be complicated functions of the complex moduli of $\Sigma$, such as $\tau, \kappa$ for (9). Much of this paper is devoted to studying different ways to define a *quantization* of these periods.

**Remark 1.** *A difference with standard 4d $\mathcal{N} = 2$ Seiberg-Witten descriptions is in the relation between BPS charges and homology classes of cycles on $\Sigma$. A careful analysis of BPS cycles [41] reveals that the logarithmic structure of the Seiberg-Witten differential for 4d KK theories plays an important role in the computation of central charges, and it imposes certain selection rules on true BPS charges. Precisely, BPS cycles are paths on $\Sigma$ that lift to closed cycles on $\tilde{\Sigma}$, a covering of $\Sigma$ induced by the logarithmic map $e^y \to y$.*

The charge lattice of BPS states for the $E_1$ theory is generated by four cycles on the mirror curve [42]

$$\Gamma = \bigoplus_{i=1}^{4} \gamma_i \mathbb{Z}. \tag{12}$$

Mirror symmetry relates $\gamma_i$ to charges of $B$-branes on $X = O_{\mathbb{P}^1 \times \mathbb{P}^1}(-2, -2)$

$$\gamma_1 : \mathcal{O}(0,0), \qquad \gamma_2 : \mathcal{O}(1,0), \qquad \gamma_3 : \mathcal{O}(1,1), \qquad \gamma_4 : \mathcal{O}(2,1), \tag{13}$$

see [43, Example 6.5(b)]. In the language of type IIA D-branes wrapping cycles on the toric Calabi-Yau, these translate into

$$\gamma_1 : D4, \qquad \gamma_2 : D2_f \overline{D4}, \qquad \gamma_3 : D0\,\overline{D2}_f D2_b \overline{D4}, \qquad \gamma_4 : \overline{D2}_b D4, \tag{14}$$

as explained in [42, Section 3]. Readers are referred to [42–51] and references therein for background and for more details on the case at hand. The intersection pairing of the four basis cycles is

$$\langle \gamma_i, \gamma_j \rangle = \begin{pmatrix} 0 & -2 & 0 & 2 \\ 2 & 0 & -2 & 0 \\ 0 & 2 & 0 & -2 \\ -2 & 0 & 2 & 0 \end{pmatrix}. \tag{15}$$

The two-dimensional sublattice $\Gamma_f$ of flavor charges, corresponding to the kernel of this pairing, is generated by

$$\gamma_{D0} = \gamma_1 + \gamma_2 + \gamma_3 + \gamma_4, \qquad \gamma_{D2_f \overline{D2}_b} = \gamma_2 + \gamma_4. \tag{16}$$

A special feature of flavor cycles is that their periods, which can be computed by direct integration of (10), are simple functions of complex moduli[2]

$$Z_{\gamma_{D0}} = \frac{2\pi}{R}, \qquad Z_{\gamma_{D2_f \overline{D2}_b}} = \frac{2i}{R} \log \tau. \tag{17}$$

## 2.2 BPS spectrum in a collimation chamber

The BPS spectrum of this theory has been studied from several angles and with different techniques, see [37,42,49,52] and references therein. A complete description of the BPS spectrum appeared in [38]. A connection to the Cremona group of $X$ in our previous work [39] led to exact computations for other local toric threefolds.

A fundamental role in the derivation of the BPS spectrum is played by a careful choice of stability condition. While for generic moduli the spectrum is difficult to compute, in certain regions known as 'collimation chambers' [39] the spectrum simplifies dramatically. An example of a collimation chamber for local $\mathbb{P}^1 \times \mathbb{P}^1$, first studied in [37] corresponds to the following configuration of central charges

$$\mathcal{C}_1^{(0)}: \quad Z_{\gamma_1} = Z_{\gamma_3}, \quad Z_{\gamma_2} = Z_{\gamma_4}, \quad \arg Z_{\gamma_1} > \arg Z_{\gamma_2}, \quad Z_{\gamma_1 + \gamma_2} \in \mathbb{R}^+. \tag{18}$$

For reasons that will become clear in a moment, we will denote the class of stability conditions $\mathcal{C}_1^{(0)}$ as the *fine-tuned stratum* of the collimation chamber. In fact the conditions (18) together with the known values of flavor central charges (17) determine the central charges of basis cycles entirely

$$Z_{\gamma_1} = Z_{\gamma_3} = \frac{\pi}{R} - \frac{i}{R} \log \tau, \qquad Z_{\gamma_2} = Z_{\gamma_4} = \frac{i}{R} \log \tau. \tag{19}$$

In particular, from (14) it follows that

$$Z_{\gamma_1 + \gamma_2} = Z_{\gamma_3 + \gamma_4} = Z_{\gamma_{D2_f}} = \frac{1}{2} Z_{\gamma_{D0}} = \frac{\pi}{R} \in \mathbb{R}^+, \tag{20}$$

where $Z_{D0} = \frac{2\pi}{R}$ is our choice of normalization for the D0 brane central charge.[3] Working with a fixed radius $R$, we may parameterize the locus $\mathcal{C}_1^{(0)}$ entirely by the value of $Z_{\gamma_1} \in \mathbb{C}$

$$Z_{\gamma_1} = Z_{\gamma_3}, \qquad Z_{\gamma_2} = Z_{\gamma_4} = \frac{\pi}{R} - Z_{\gamma_1}, \tag{21}$$

with $0 \le \operatorname{Re} Z_{\gamma_1} < \pi/R$ and $\operatorname{Im} Z_{\gamma_1} > 0$, so that all $Z_{\gamma_i}$ are contained in the half-plane with phases $-\pi/2 < \arg Z_{\gamma_i} \le \pi/2$.

In the context of 4d $\mathcal{N} = 2$ QFT and supergravity, the BPS spectrum is encoded by the BPS index $\Omega(\gamma) \in \mathbb{Z}$ (a.k.a. the 'second helicity supertrace'). The notion of BPS index can be

---

[2]The relevant cycles can be found in [42]. Below we will review a computation for $Z_{D0} = 2\pi/R$ in the half-geometry that is similar to the one for $Z_{\gamma_{D0}}$ in (9). To compute $Z_{\gamma_{D2_f \overline{D2}_b}}$ observe that the sum of these cycles, obtained by lifting saddles $p_5 - p_4$ in [42, Figure 5], corresponds to small loops near the punctures at $e^x = e^y = 0$ and $e^x = e^y = \infty$. Near these punctures the differential becomes $2\pi R \lambda = y\, dx \sim \log \tau\, dz/z + d(\log z)^2$, in coordinate $z = e^x$. This has a simple pole at the puncture $z = 0$ (similarly near $z = \infty$), with residue $-2\pi i \log \tau$. Summing up the two contributions gives the claimed result.

[3]This is consistent with (10) and (11) as shown by direct computation of the D0 brane period in [41].

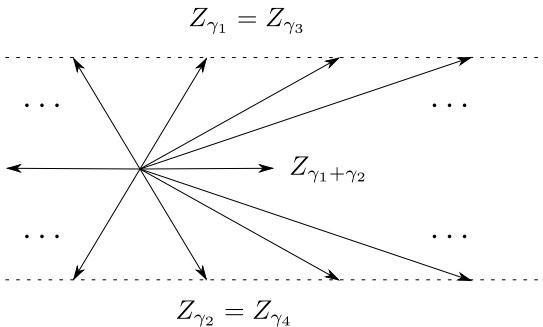

Figure 2: Part of the BPS spectrum (22), for stability condition (18).

extended to 5d $\mathcal{N} = 1$ QFT on the circle, viewed as a 4d $\mathcal{N} = 2$ theories of the Kaluza-Klein modes. The BPS spectrum in $\mathcal{C}_1^{(0)}$ is[4]

$$
\begin{aligned}
\Omega(\gamma_1 + k(\gamma_1 + \gamma_2)) = \Omega(\gamma_3 + k(\gamma_3 + \gamma_4)) &= 1 \,, \\
\Omega(\gamma_1 + \gamma_2 + k\gamma_{D0}) = -2 \,, \quad k &\in \mathbb{Z} \,,
\end{aligned}
\tag{22}
$$

together with CPT conjugates with $\Omega(-\gamma) = \Omega(\gamma)$. There is a second collimation chamber $\mathcal{C}_2$, obtained by setting $\arg Z_{\gamma_1} < \arg Z_{\gamma_2}$ in (18), where the spectrum takes the form (22) after cyclic permutation $(1, 2, 3, 4)$ of the charge labels [39].

### 2.2.1 Geometric realization of the fine-tuned stratum

It is natural to ask whether the stability condition (18) is actually present in the physical moduli space of the theory, corresponding to the complex moduli space of the curve (9) parameterized by $\tau, \kappa$. If this is the case, it follows that the BPS spectrum (22) is actually realized in the physical theory, namely 5d $\mathcal{N} = 1$ $SU(2)$ Yang-Mills. Otherwise the BPS spectrum would be unphysical, although it could still be used to compute the wall-crossing invariant of [55] to deduce the physical spectrum for other stability conditions. At the time when the spectrum was studied by [38, 39, 56] the answer to this question was not known. Here we settle the question in the affirmative.

We will now show that the fine-tuned stability condition (18) is realized by periods of $\Sigma$ in the region of moduli space

$$
\kappa \to 0 \,, \quad \text{with} \quad \tau \text{ finite} \,.
\tag{23}
$$

Appendix A contains a detailed analysis of the mirror curve, its periods, and some of the basic BPS states from exponential networks.

In the fine-tuned limit (23) the curve becomes

$$
\tau(e^x + e^{-x}) + e^y + e^{-y} = 0 \,.
\tag{24}
$$

Away from the punctures, the curve is still smooth and the periods are (see Appendix A)[5]

$$
\frac{1}{2\pi R} \oint_{\gamma_1, \gamma_3} \lambda = \frac{\pi}{R} - \frac{i}{R} \log \tau \,, \qquad \frac{1}{2\pi R} \oint_{\gamma_2, \gamma_4} \lambda = \frac{i}{R} \log \tau \,,
\tag{25}
$$

---

[4]We omit here the D0 branes, whose BPS index would be $\Omega(\gamma_{D0}) = -4$. On the one hand they do not belong to the strict field theory limit [53, 54]. On the other hand they do not participate in wall-crossing, and therefore do not affect the structure of BPS chambers.

[5]We thank the anonymous referee for suggesting an elegant argument to show the symmetry of the periods.

as predicted by (19). This confirms that (18), with either $\arg Z_{\gamma_1} \gtrless \arg Z_{\gamma_2}$ (corresponding to conditions $\mathcal{C}_1^{(0)}$ or $\mathcal{C}_2^{(0)}$), is indeed realized by $Z_\gamma$ computed as periods (10). Since the moduli space of $\Sigma$ coincides with the physical moduli space of the 5d gauge theory [11], it follows that the BPS spectrum (22) is indeed physical.

The fine-tuned stratum is parameterized solely by $\tau$. Noting that

$$\text{Re}\, Z_{\gamma_1} = \frac{\pi}{R} + \frac{1}{R} \arg \tau\,, \qquad \text{Re}\, Z_{\gamma_2} = -\frac{1}{R} \arg \tau\,, \tag{26}$$

all basic central charges $Z_{\gamma_i}$ will lie in the right-half plane if

$$0 < \arg \tau < \pi\,. \tag{27}$$

Whenever this condition is violated, at least two of the basic central charges exit the half-plane. In a similar fashion, since

$$\text{Im}\, Z_{\gamma_1} = -\text{Im}\, Z_{\gamma_2} = -\frac{1}{R} \log|\tau|\,, \tag{28}$$

it follows that the fine-tuned locus is divided into two regions

$$\begin{aligned} |\tau| < 1 &\quad \Rightarrow \quad \arg Z_{\gamma_1} > \arg Z_{\gamma_2}\,, \\ |\tau| > 1 &\quad \Rightarrow \quad \arg Z_{\gamma_1} < \arg Z_{\gamma_2}\,. \end{aligned} \tag{29}$$

These regions correspond to the fine-tuned loci $\mathcal{C}_1^{(0)}$ and $\mathcal{C}_2^{(0)}$ described earlier. The two regions have different BPS spectra, therefore $|\tau| = 1$ corresponds to a wall of marginal stability.

The region $\mathcal{C}_1^{(0)}$ corresponding to $|\tau| < 1$ includes the distinguished point $\tau = 0$, which corresponds to a degeneration of the mirror curve into two half-geometries (see Section 2.3). Likewise the region $\mathcal{C}_2$ corresponding to $|\tau| > 1$ includes the distinguished point $\tau = \infty$, which corresponds to a different degeneration into two half-geometries. This is summarized in Figure 3.

### 2.2.2 An affine symmetry on the fine-tuned stratum

It will be useful to observe that there is a distinguished $\mathbb{Z}$-action on each of the the chambers $\mathcal{C}_i^{(0)}$. For example in $\mathcal{C}_1^{(0)}$, this is the rotation by $\pi$

$$T(\tau) = e^{\pi i} \tau\,, \tag{30}$$

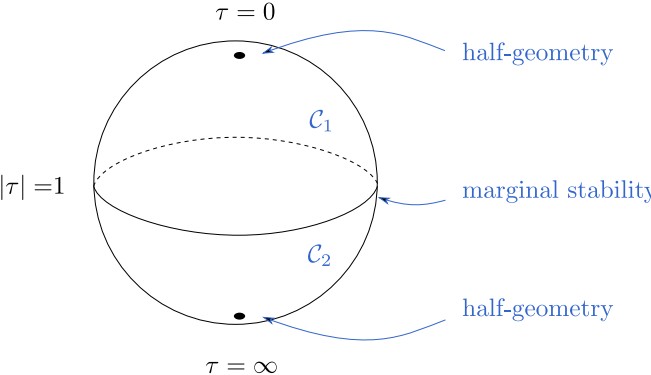

Figure 3: The fine tuned stratum. Two BPS chambers $\mathcal{C}_1^{(0)}, \mathcal{C}_2$ separated by a wall of marginal stability $|\tau| = 1$. Each chamber includes a half-geometry point.

acting as follows on the central charges:

$$T: \quad \begin{aligned} &Z_{\gamma_1} \to Z_{\gamma_1+(\gamma_1+\gamma_2)}, \quad Z_{\gamma_2} \to Z_{\gamma_2-(\gamma_1+\gamma_2)}, \\ &Z_{\gamma_3} \to Z_{\gamma_3+(\gamma_3+\gamma_4)}, \quad Z_{\gamma_4} \to Z_{\gamma_4-(\gamma_3+\gamma_4)}. \end{aligned} \tag{31}$$

Combining this with a relabeling of charges

$$T: \quad \begin{aligned} &\gamma_1 \to \gamma_1+(\gamma_1+\gamma_2), \quad \gamma_2 \to \gamma_2-(\gamma_1+\gamma_2), \\ &\gamma_3 \to \gamma_3+(\gamma_3+\gamma_4), \quad \gamma_4 \to \gamma_4-(\gamma_3+\gamma_4), \end{aligned} \tag{32}$$

we find a symmetry of the BPS spectrum (22). By this we mean that central charges, BPS indices, and Dirac pairings of the spectrum obtained by acting with $T$ are identical to those of the original spectrum

$$T(Z_\gamma) = Z_{T(\gamma)}, \qquad \Omega(T(\gamma)) = \Omega(\gamma), \qquad \langle \gamma, \gamma' \rangle = \langle T(\gamma), T(\gamma') \rangle. \tag{33}$$

A distinguishing feature of the relabeling (32) is that it coincides with the one arising from a sequence quiver mutation, or equivalently from a 'tilting' of the positive half-plane [39, 52]. Indeed, the $\mathbb{Z}$-action (30) shifts the basis central charges by $\pm\frac{\pi}{R}$, which pushes two of them outside of the right half-plane, according to (27). This induces a change in the quiver description, precisely by a pair of mutations.

### 2.2.3 Away from the fine-tuned stratum

As it turns out, stability conditions (18) are rather peculiar. For example, we will see in Section 4, that the associated Riemann-Hilbert problem in the sense of [1, 57] becomes trivial in the 'conformal limit', in spite of the fact that the system is coupled. It may be observed however that the fine-tuned stratum (18) is only part of a larger chamber in the moduli space of stability conditions.

Here we will define two one-parameter families of stability conditions that deform the fine-tuned stratum (18). The first one is

$$\mathcal{C}_1^{(\delta)}: \quad Z_{\gamma_3} = Z_{\gamma_1} + \delta, \quad Z_{\gamma_4} = Z_{\gamma_2} - \delta, \quad \arg Z_{\gamma_1} > \arg Z_{\gamma_2}, \quad Z_{\gamma_1+\gamma_2} = \frac{\pi}{R} \in \mathbb{R}^+, \tag{34}$$

with

$$-\frac{\pi}{R} < \delta < \frac{\pi}{R}. \tag{35}$$

It still satisfies (20) and belongs to the same chamber as $\mathcal{C}_1^{(0)}$, as we now explain.[6]

On the one hand the limiting rays ($k \to \pm\infty$) for the spectrum (22) still lie on the real axis, since the relation (20) is unchanged. On the other hand, observe from (22) that BPS states with central charges (18) are arranged according to two identical and overlapping 'peacock patterns' [58, 59], as shown in Figure 2. The presence of coincident rays in the pattern is allowed because the corresponding charges are mutually local, namely

$$\langle \gamma_1 + k(\gamma_1 + \gamma_2), \gamma_3 + k(\gamma_3 + \gamma_4) \rangle = 0. \tag{36}$$

Turning on the deformation $\delta$ resolves the two patterns as shown in Figure 4. In order to remain within the same chamber it is crucial that no rays in the complex plane of central charges cross each other, except for mutually local ones. In particular we should avoid a crossing between any pair of charges with

$$\langle \gamma_1 + k(\gamma_1 + \gamma_2), \gamma_3 + k'(\gamma_3 + \gamma_4) \rangle \neq 0, \qquad \text{if } k \neq k'. \tag{37}$$

---

[6]Depending on $\delta$, a small tilt of the half-plane may be necessary so that it contains all four $Z_{\gamma_i}$ at once.

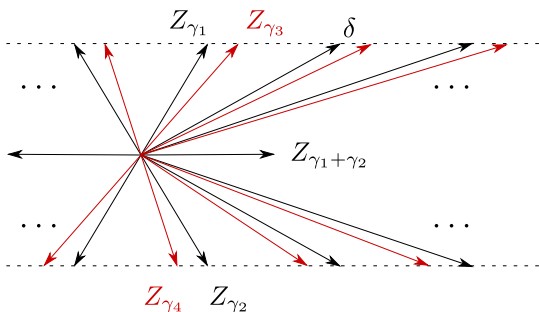

Figure 4: Part of the BPS spectrum (22), for $\delta$-deformed stability condition (34).

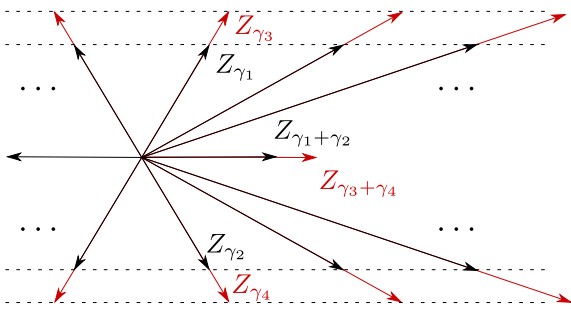

Figure 5: Part of the BPS spectrum (22), for $\rho$-deformed stability condition (39).

This is ensured by the condition (35), since the spacing between two central charges in the sequence $\gamma_1 + k(\gamma_1 + \gamma_2)$ is given precisely by $Z_{\gamma_1+\gamma_2} = \pi/R$. The class of stability conditions (34) can be parameterized by the complex number $Z_{\gamma_1}$ subject to the same constraints as before, and by the real $\delta$ subject to (35).

Thanks to the periodic 'peacock' pattern characterizing the BPS spectrum, it is possible to extend the $\mathbb{Z}$-action (30) defined on the fine-tuned stratum to the more general class of stability conditions $\mathcal{C}_1^{(\delta)}$. Taking

$$T(Z_{\gamma_{1,3}}) = Z_{\gamma_{1,3}} + \frac{\pi}{R}, \qquad T(Z_{\gamma_{2,4}}) = Z_{\gamma_{2,4}} - \frac{\pi}{R}, \tag{38}$$

preserves both (34) and (35). As before, this $\mathbb{Z}$-action pushes two of the basic BPS states (those with charges $\gamma_i$) to exit the right half-plane. From the viewpoint of a quiver description based on the right half-plane, this induces a pair of mutations, as will be seen in more detail in Section 5.

The second deformation is

$$\mathcal{C}_1^{(\rho)}: \quad Z_{\gamma_3} = \rho Z_{\gamma_1}, \qquad Z_{\gamma_4} = \rho Z_{\gamma_2}, \qquad \arg Z_{\gamma_1} > \arg Z_{\gamma_2}, \qquad \rho, Z_{\gamma_1+\gamma_2}, \in \mathbb{R}^+. \tag{39}$$

This is still trivially in the same chamber of $\mathcal{C}_1^{(0)}$, since we have not changed the phase of any central charge, as the only bound states away from the real axis occur only between $\gamma_1$ and $\gamma_2$ or $\gamma_3$ and $\gamma_4$. The BPS spectrum is then unchanged, but organized in two parallel peacock patterns instead of one, as shown in Figure 5, and the $\mathbb{Z}$-action (30) can be extended in a similar manner as before.

## 2.3 Half-geometry limit

We conclude this section by studying the limit where local $\mathbb{P}^1 \times \mathbb{P}^1$ degenerates into its 'half-geometry'

$$O_{\mathbb{P}^1 \times \mathbb{P}^1}(-2,-2) \qquad \to \qquad O_{\mathbb{P}^1}(-2) \oplus O_{\mathbb{P}^1}(0). \tag{40}$$

One motivation for considering this limit is the observation [38] that a class of stability conditions very close to (34) can be realized by considering the limit $\tau \to 0$ corresponding to the degeneration (40).[7] A second motivation for studying the half-geometry is that taking $\tau \to 0$ corresponds to a weak-coupling limit for 5d $\mathcal{N} = 1$ $SU(2)$ Yang-Mills theory [11]. This observation will be important in connection to a description of quantum periods in terms of instanton partition functions, whose computation takes place in the weak-coupling regime of the gauge theory [60, 61], and will be the focus of section 5.

Taking $\tau \to 0$ while staying in the physical slice of the collimation chamber $\mathcal{C}_1$, from (17) we expect

$$Z_{\gamma_1} \approx Z_{\gamma_3} \to +i\infty, \qquad Z_{\gamma_2} \approx Z_{\gamma_4} \to -i\infty. \tag{41}$$

Geometrically this reflects the fact that cycles corresponding to the $D4$ brane and the $D2_b$ brane grow to infinite size. In terms of the stability condition $\mathcal{C}_1^\delta$ in (34), we may reproduce this by keeping $\delta$ fixed while taking $\arg Z_{\gamma_1} = \pi/2 - \epsilon$ with $\epsilon \to 0^+$. In this limit

$$Z_{\gamma_1+\gamma_2} = Z_{\gamma_{D2_f}} = \frac{1}{2} Z_{\gamma_{D0}} = \frac{\pi}{R}, \qquad Z_{\gamma_3+\gamma_4} = Z_{D0\overline{D2_f}} = \frac{1}{2} Z_{\gamma_{D0}} = \frac{\pi}{R} \tag{42}$$

remain fixed, so that the stability condition always belongs to $\mathcal{C}_1^{(\delta)}$. Therefore the BPS spectrum cannot change by wall-crossing, but can at most shed part of its states when reaching the boundaries of the chamber as $\tau \to 0$. Indeed it was observed that in this limit all states that become infinitely massive disappear, while all states that retain finite mass get their BPS index halved [42]. The states that survive are $D2\,D0$ boundstates with charges[8]

$$\Omega(\gamma_1 + \gamma_2 + k\gamma_{D0}) = -1, \qquad k \in \mathbb{Z}, \tag{43}$$

together with the CPT conjugates (see [62, Section 4]). Note that this spectrum is a subset of that in (22) corresponding to chamber $\mathcal{C}_1$.

We may also check that the expectation (42) on the behaviour of periods in the limit $\tau \to 0$ is indeed verified by the mirror geometry. A simple computation, involving a change of variables described in Appendix B leads to the following mirror curve for the half-geometry

$$(1+Q) - e^x - Q\,e^{-y} - e^y = 0, \tag{44}$$

where $Q$ is related to $\kappa$ by

$$\kappa^2 = \frac{(1+Q)^2}{Q}. \tag{45}$$

In section 3 we will show that the quantum periods of this curve are classically exact and given by

$$Z_{\gamma_{D2_f}} = -\frac{i}{R} \log Q, \qquad Z_{\gamma_{D0}} = \frac{2\pi}{R}, \tag{46}$$

where $\gamma_{D2_f}$ is the cycle corresponding to the limit of $\gamma_1 + \gamma_2$ in the full geometry. Since these periods match exactly the periods in the fine-tuned stratum (42) for $Q = e^{\pi i}$, we conclude that the collimation chamber $\mathcal{C}_1$ contains at least one point where local $\mathbb{P}^1 \times \mathbb{P}^1$ degenerates to a

---

[7]In [38, 42] the curve is parameterized by $Q_b, Q_f$ which are related to our moduli as $\tau = Q_b/Q_f$ and $\kappa = \kappa$. The limit considered in the references is therefore $Q_b \to 0$.

[8]Again we are neglecting contributions from pure $D0$ branes, which also get halved.

half-geometry. Indeed, since taking $Q \to e^{\pi i}$ sends $\kappa \to 0$, this limit may also be regarded as a special case of (23) where we also take $\tau \to 0$.

It is worth mentioning that one may also take a limit on the space of stability conditions along $\mathcal{C}_1^{(\rho)}$, in which we take again (41) but with (42) replaced by

$$Z_{\gamma_1 + \gamma_2} = Z_{\gamma_{D2_f}} = \frac{Z_{\gamma_{D0}}}{1 + \rho} = \frac{2\pi}{R(1 + \rho)}, \qquad Z_{\gamma_3 + \gamma_4} = Z_{D0 \overline{D2}_f} = \frac{\rho}{1 + \rho} Z_{\gamma_{D0}} = \frac{2\pi\rho}{R(1 + \rho)}. \quad (47)$$

Regardless of whether $\mathcal{C}_1^{(\rho)}$ belongs to the physical slice of the collimation chamber, the limiting configuration of central charges (47) belongs to the physical moduli space of the half-geometry, with the identification

$$Q = e^{\frac{2\pi i}{1 + \rho}}. \quad (48)$$

Therefore, the $\rho$-deformation, unlike the $\delta$-deformation, allows us to explore the imaginary direction in the parameter space of the half-geometry.

## 3 Quantum periods from WKB

The definition of quantum periods is based on a $\hbar$-difference equation ($\hbar$DE) associated to the classical (mirror) curve $\Sigma$, by imposing the WKB ansatz

$$\psi(x; \hbar) = \exp\left(\int^x S(x; \hbar) dx\right), \qquad S(x; \hbar) = \frac{2\pi R}{\hbar} \lambda + O(1), \quad (49)$$

where $S(x; \hbar)$ is an asymptotic power series in $\hbar$, and $\lambda$ is the classical differential (11). Monodromies of $\psi$ are described by exponentiated contour integrals of $S(x; \hbar) dx$, which are also asymptotic series in $\hbar$. Under the assumption that the series are Borel summable, the quantum periods are defined to be the Borel summation of the WKB periods[9]

$$\Pi_\gamma(\hbar) := \mathcal{B}\left[\oint_\gamma S(x; \hbar) dx\right] = \frac{2\pi R}{\hbar} Z_\gamma + O(1). \quad (50)$$

For later convenience we will denote the exponentiated quantum periods by

$$X_\gamma := \exp \Pi_\gamma. \quad (51)$$

### 3.1 $\hbar$-difference equations

The quantization of mirror curves in the context of refined open topological strings gives rise to $\hbar$-difference equations [66].[10] The open string partition function plays the role of a wavefunction [67–71], characterized by the $\hbar$-difference equation and by a certain choice of boundary conditions. In the following we will consider quantum curves that appear in the study of open refined Topological Strings in the Nekrasov-Shatashvili limit [22, 23, 72]. Given an algebraic curve $\Sigma$

$$F(e^x, e^y) = \sum_{m,n} a_{m,n} e^{mx} e^{ny} = 0, \quad (52)$$

---

[9]The definition of Borel sum can be found in many standard references, see e.g. [63]. We follow closely the conventions of [64]. For discussions of Borel summability in the context of first-order $\hbar$-difference equations, we refer to the recent papers [28, 29, 65].

[10]These are also known as $q$-difference equations in the literature, where the relevant equations are often expressed in terms of of exponentiated variables. We slightly deviate from standard jargon in order to preserve $q$ for q-Painlevé equations, whose parameter $q$ is slightly different from the one that would appear in $q$-difference equations arising from quantum mirror curves.

in variables $(e^x, e^y) \in \mathbb{C}^* \times \mathbb{C}^*$, the corresponding quantum curve is a $\hbar$-difference equation

$$\hat{F}(e^{\hat{x}}, e^{\hat{y}})\psi(x) = \left(\sum_{m,n} a_{m,n}(\hbar)e^{m\hat{x}}e^{n\hat{y}}\right)\psi(x) = 0, \tag{53}$$

with

$$\lim_{\hbar \to 0} a_{m,n}(\hbar) = a_{m,n}, \qquad e^{\hat{x}}\psi(x) = e^x\psi(x), \qquad e^{\hat{y}}\psi(x) = \psi(x + \hbar). \tag{54}$$

Here $\psi(x)$ is truly a function of $e^x$, meaning $\psi(x + 2\pi i) = \psi(x)$. This definition of a quantum curve involves a choice of polarization based on classical coordinates $(x, y)$ whose quantization is achieved by replacing

$$y \to \hat{y} = \hbar \partial_x. \tag{55}$$

It will later be useful to consider other choices of polarization, which can be obtained by an $Sp(2, \mathbb{Z})$ transformation on $(x, y)$.

Clearly, there is more than one $\hbar$-difference equation that reduces to a given algebraic curve $\Sigma$. This ambiguity may be traced to the $\hbar$-dependence of $a_{m,n}(\hbar)$, which trivializes in the limit $\hbar \to 0$, and has to be fixed by some additional requirements. For example, a popular convention, known as Weyl's prescription [73–75], consists in replacing $e^{mx+ny}$ by $e^{m\hat{x}+n\hat{y}}$. With this prescription the curve (52) is promoted to the following $\hbar$-difference equation

$$\hat{F}(e^{\hat{x}}, e^{\hat{y}})\psi(x) = \left(\sum_{m,n} a_{m,n} e^{m\hat{x}+n\hat{y}}\right)\psi(x) = 0, \tag{56}$$

where $a_{m,n}$ are the classical coefficients.

## 3.2 Quantum periods

We will now show that the quantum periods (50) can be written in terms of the eigenvalue $\mathcal{R}$ of the shift operator $e^{\hat{y}}$,

$$\mathcal{R}(x; \hbar) := \frac{\psi(x + \hbar)}{\psi(x)}. \tag{57}$$

Left-multiplying (53) by $[\psi(x)]^{-1}$ and taking the limit $\hbar \to 0$, it is clear that

$$\lim_{\hbar \to 0} \mathcal{R}(x; \hbar) = \exp y(x), \tag{58}$$

where $y(x)$ is a sheet of the classical curve (52). It follows that, while $\mathcal{R}(x; \hbar)$ is generally a multi-valued function of $x$, its (semi-) classical limit is single valued on $\Sigma$. From (58) it follows that the leading order of $\frac{1}{2\pi R}\log \mathcal{R}$ coincides with the classical differential $\lambda$ (11), so that

$$\frac{1}{\hbar}\oint_\gamma \log \mathcal{R}(x; \hbar)\, dx = \frac{2\pi R}{\hbar} Z_\gamma + O(\hbar^0). \tag{59}$$

In fact, it is possible to write $\mathcal{R}$ in terms of $S$: from (49) and (57) it follows that

$$\mathcal{R}(x, \hbar) = \exp\left\{\int_x^{x+\hbar} dx\, S(x; \hbar)\right\} = \exp\left\{\hbar S(x; \hbar) + \sum_{k=1}^\infty \frac{\hbar^{k+1}}{(k+1)!}\partial_x^k S(x; \hbar)\right\}, \tag{60}$$

which means that $\frac{1}{\hbar}\log \mathcal{R}(x)dx$ and $S(x)dx$ only differ by a total derivative

$$S(x; \hbar)\, dx = \frac{1}{\hbar}\log \mathcal{R}(x; \hbar)dx + d\xi(x; \hbar), \qquad \xi(x; \hbar) := \sum_{k=1}^\infty \frac{\hbar^{k+1}}{(k+1)!}\partial_x^{k-1} S(x; \hbar). \tag{61}$$

Therefore periods of $S(x;\hbar)dx$ along closed BPS cycles $\gamma$ actually coincide with the periods of $\frac{1}{\hbar}\log\mathcal{R}(x;\hbar)dx$

$$\frac{1}{\hbar}\oint_\gamma \log\mathcal{R}(x;\hbar)\,dx = \int_\gamma S(x;\hbar),\tag{62}$$

and we can compute the quantum periods (50) using the 1-form $\log\mathcal{R}\,dx$ instead. This will turn out to be important, as we will see in the following sections that $\mathcal{R}$ is the solution to a $\hbar$-difference version of the Riccati equation, and its $\hbar$-expansion can be systematically computed. In the following, we will refer to $\frac{1}{\hbar}\log\mathcal{R}(x;\hbar)dx$ as the quantum, or WKB, differential.

**Remark 2.** *Equation* (62) *holds if the function $\xi$ in equation* (61) *is single-valued on the logarithmic cover $\tilde{\Sigma}$ of $\Sigma$ used to define BPS cycles. It follows from the definition* (49) *of $S\,dx$ as a differential on $\tilde{\Sigma}$ that $S\,dx$ and its derivatives are single-valued on the logarithmic cover.*

### 3.3 Boundary conditions for $\psi$

The quantum one-form $\log\mathcal{R}(x,\hbar)$ defines $\psi(x)$ in terms of transport by finite shifts through (57). Iterating such shifts infinitely many times leads to an explicit solution for $\psi(x)$ in terms of $\mathcal{R}(x;\hbar)$. We have two distinct cases, depending on the sign of $\operatorname{Re}\hbar$

$$\psi(x)=\psi_0(x)\prod_{k=-\infty}^{-1}\mathcal{R}(x+k\hbar;\hbar)=\psi_\infty(x)\prod_{k=0}^{\infty}\frac{1}{\mathcal{R}(x+k\hbar;\hbar)}\qquad(\operatorname{Re}\hbar>0),\tag{63}$$

$$\psi(x)=\psi_\infty(x)\prod_{k=-\infty}^{-1}\mathcal{R}(x+k\hbar;\hbar)=\psi_0(x)\prod_{k=0}^{\infty}\frac{1}{\mathcal{R}(x+k\hbar;\hbar)}\qquad(\operatorname{Re}\hbar<0).\tag{64}$$

Here the functions $\psi_0,\psi_\infty$ are $\hbar$-periodic functions

$$\psi_0(x+\hbar)=\psi_0(x),\qquad \psi_\infty(x+\hbar)=\psi_\infty(x).\tag{65}$$

Due to linearity of the $\hbar$-difference equation (53), any solution $\psi(x)$ is ambiguously defined up to multiplication by such $\hbar$-periodic functions. This is the $\hbar$-difference uplift of the familiar statement that solutions to linear ODEs are defined up to an overall constant multiplier.

The functions $\psi_0(x),\psi_\infty(x)$ can be fixed in part by studying the boundary conditions for the wavefunction. Recall from (58) that the semiclassical limit of $\psi$ depends on a choice of branch $y(x)$ for $\Sigma$, and different branches give rise to different boundary conditions at leading order. Through the consistency imposed by (53) this dependence determines the boundary condition to higher orders in $\hbar$ as well. For instance, in the context of open topological string theory, one is often (though not always) interested in a branch where $\psi(x)=\sum_{k\geq0}\psi_k(\hbar)e^{kx}$ with $\psi_0=1$, meaning that the wavefunction has asymptotics normalized to $\psi(-\infty)=1$.

On the other hand, boundary conditions do not entirely fix the ambiguity. There may be $\hbar$-periodic factors that simply obey the desired boundary condition and one may choose to include them or not in the answer for $\psi(x)$. A prescription to fix these residual factors is to take an asymptotic expansion of $\psi(x)$ as a series in $\hbar$, and perform a Borel resummation. This resummation involves the choice of an angular sector $\measuredangle$ in the Borel plane, bounded by rays corresponding to singularities of the Borel transform of the series. Different sectors give rise to resummations with different $\hbar$-periodic normalizations

$$\mathcal{B}_{\measuredangle}[\psi(x)]=\mathcal{S}_{\measuredangle,\measuredangle'}(x)\mathcal{B}_{\measuredangle'}[\psi(x)].\tag{66}$$

A detailed analysis of this phenomenon for first-order $\hbar$-difference equations associated to the mirror of $\mathbb{C}^3$ can be found in [28]. The quantum mirror curve of the resolved conifold is discussed in [28, 29]. The choice of a normalization for $\psi(x)$ matters when solving the $\hbar$-difference equation, but does not affect the definition (50) of the quantum periods (nevertheless, quantum periods are sensitive to the phase of $\hbar$ through Stokes automorphisms). For this reason we will mostly neglect this issue in the following.

### 3.4 WKB for first order $\hbar$-difference equations

The most general first order linear $\hbar$-difference equation is simply the definition of $\mathcal{R}$:

$$\left[e^{\hat{y}} - \mathcal{R}(x,Q;\hbar)\right]\psi(x) = \psi(x+\hbar) - \mathcal{R}(x,Q;\hbar)\psi(x) = 0, \qquad (67)$$

with $\mathcal{R}$ given explicitly by the equation. As such, a solution is given straightforwardly by (63), (64). In Appendix C we discuss the examples of $\mathbb{C}^3$ and the resolved conifold; here we will focus on the half-geometry, that we saw arising as the limit $\tau \to 0$ of local $\mathbb{P}^1 \times \mathbb{P}^1$.

**Half-geometry**

The mirror curve (44) for the half-geometry can be written equivalently as

$$e^x = (1 - Qe^{-y})(1 - e^y), \qquad (68)$$

and in the limit $Q \to 0$ it degenerates to the curve of $\mathbb{C}^3$, see (C.1). We may quantize (68) by simply replacing $x, y$ with the corresponding operators $\hat{x}, \hat{y}$, but the usual polarization (55) leads to a second-order $\hbar$-difference equation, namely

$$(e^x - 1 - Q)\psi(x) + Q\psi(x - \hbar) + \psi(x + \hbar) = 0. \qquad (69)$$

We can however switch to a Fourier-dual polarization where $\hat{x} = -\hbar\partial_y$, $\hat{y} = y$, in which the $\hbar$-difference equation becomes

$$\tilde{\psi}(y - \hbar) = (1 - Qe^{-y})(1 - e^y)\tilde{\psi}(y), \qquad (70)$$

which is a first-order system for the Fourier transform $\tilde{\psi}$ of $\psi$. We can easily find a solution with boundary condition $\tilde{\psi}(-\infty) = 1$

$$\tilde{\psi}(y) = \frac{(Qe^{y+\hbar}; e^{\hbar})_\infty}{(e^{-y}; e^{\hbar})_\infty}. \qquad (71)$$

To compute the parallel transport for the Fourier-dual wavefunction we define

$$\tilde{\mathcal{R}}(y;\hbar) = \frac{1}{\tilde{\psi}(y)} x \cdot \tilde{\psi}(y) = \frac{\tilde{\psi}(y - \hbar)}{\tilde{\psi}(y)} = (1 - Qe^{-y})(1 - e^y). \qquad (72)$$

Repeating the arguments that led to (62) one may show that $-\log\tilde{\mathcal{R}}\, dy$ is in the same cohomology class as $d\log\tilde{\psi}$. Since quantum periods are invariant under changes of polarization,[11] they can be computed from $\tilde{\mathcal{R}}$ as follows

$$\Pi_\gamma = -\frac{1}{\hbar}\oint_\gamma \log\tilde{\mathcal{R}}(y;\hbar)\, dy, \qquad (73)$$

by computing the primitive

$$-\frac{1}{\hbar}\int^y \log\tilde{\mathcal{R}}(y)\, dy = \frac{1}{\hbar}\left[\text{Li}_2(Q^{-1}e^y) + \text{Li}_2(e^y) - \frac{\pi^2}{6} - \frac{1}{2}\log^2(-Qe^{-y})\right], \qquad (74)$$

---

[11]A slick argument for this, is via the correspondence between quantum periods of exact WKB with solutions of TBA equations. The latter do not involve a choice of polarization for the quantum curve (in fact they do not even involve the choice of a quantum curve). A more direct derivation of this statement involves passing from a wavefunction $\psi$ in one polarization to another polarization via Fourier transform $\tilde{\psi}$. By direct inspection, it is not hard to see that quantum periods obtained by transport of $\psi$ coincide with those of $\tilde{\psi}$.

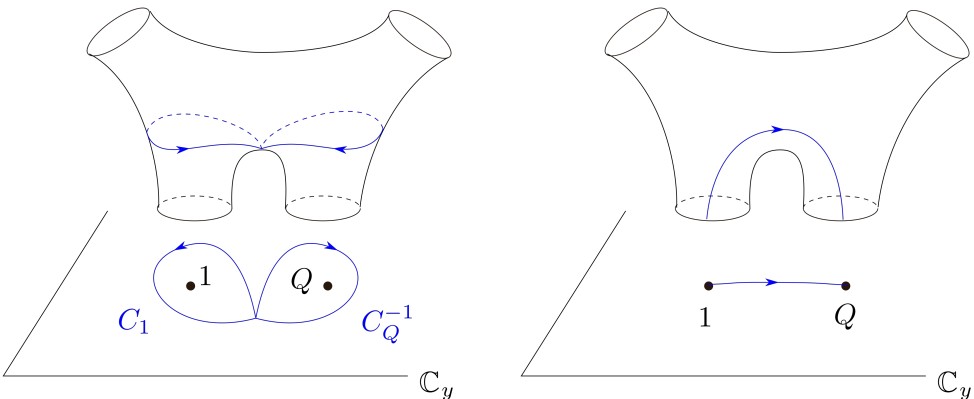

Figure 6: Left: the cycle $\gamma_{D2}$ on the mirror curve of the half-geometry, shown as a covering over the $y$ plane. Labels of punctures denote the values of $e^y = 1, Q$ respectively. Right: the cycle $\gamma_{D4}$ in the half-geometry limit becomes noncompact.

and then studying its monodromies along appropriate cycles of $\Sigma$. The mirror curve (68) is a sphere with four punctures, located at $e^y = 0, 1, Q$, with two independent periods.[12] Let us denote by $C_z$ a small counter-clockwise loop around the puncture at $e^y = z$. In [62] via exponential networks, the cycle corresponding to the $D2$ brane in the mirror picture is found to be

$$\gamma_{D2} = C_Q^{-1} \circ C_1, \tag{75}$$

depicted in Figure 6. Using the monodromy properties (C.7), we find that

$$
\begin{aligned}
\mathrm{Li}_2(Q^{-1}e^y) + \mathrm{Li}_2(e^y) &\xrightarrow{C_1} \mathrm{Li}_2(Q^{-1}e^y) + \mathrm{Li}_2(e^y) - 2\pi i \log e^y \\
&\xrightarrow{C_Q^{-1}} \mathrm{Li}_2(Q^{-1}e^y) + 2\pi i \log(Q^{-1}e^y) + \mathrm{Li}_2(e^y) - 2\pi i \log e^y \\
&= \mathrm{Li}_2(Q^{-1}e^y) + \mathrm{Li}_2(e^y) - 2\pi i \log Q,
\end{aligned}
\tag{76}
$$

so that the quantum period is

$$\Pi_{\gamma_{D2}} = -\frac{2\pi i}{\hbar} \log Q = \frac{2\pi R}{\hbar} Z_{\gamma_{D2}}. \tag{77}$$

To complete the basis of quantum periods we need a second, linearly independent, cycle. We choose the cycle $\gamma_{D0}$ corresponding to the $D0$-brane, whose computation is completely analogous and is performed in Appendix C for the resolved conifold. The idea is to note that (68) is a pair of trinions glued along a tube, and then recall that each trinion is a copy of the mirror curve of $\mathbb{C}^3$. The D0 cycle can be embedded into, say, the left trinion: if $C_0, C_1, C_\infty$ are cycles around the three punctures of the trinion, then

$$\gamma_{D0} = C_1^{-1} \circ C_0^{-1} \circ C_\infty^{-1} = C_1^{-1} \circ C_0^{-1} \circ C_1 \circ C_0, \tag{78}$$

also see [62, Figure 18]. The period computation proceeds by keeping track of monodromies of the primitive, and turns out to be

$$\Pi_{\gamma_{D0}}(\hbar) = \frac{4\pi^2}{\hbar} = \frac{2\pi R}{\hbar} Z_{\gamma_{D0}}, \tag{79}$$

---

[12]The number of independent periods is not $b_1(\Sigma)$ due in part to logarithmic branching of $\lambda$. For details, see [41, 62].

A similar computation for the D0 in the resolved conifold is detailed in Appendix C, see equation (C.15). Note that the periods of first-order systems are almost trivially computed, since the only possible $\hbar$-dependence of the quantum differential $\log\mathcal{R}$ is the one explicitly coming from the first-order equation (67) (this explicit dependence is absent for the half-geometry quantum curve (69)).

## 3.5 WKB for second order $\hbar$-difference equations

Higher-order $\hbar$-difference equations arise typically for Calabi-Yau geometries that engineer interacting five-dimensional theories.[13] We focus on second-order $\hbar$-difference equations,[14] that can generally be written as

$$\psi(x+2\hbar)+a_1(x;\hbar)\psi(x+\hbar)+a_2(x;\hbar)\psi(x)=0\,. \tag{80}$$

Here $a_i$ are really functions of $e^x$, in other words they obey the periodicity constraints

$$a_i(x;\hbar)=a_i(x+2\pi i;\hbar)\,. \tag{81}$$

Generically they may also depend on $\hbar$ and on the complex moduli of $\Sigma$. We will assume for simplicity that they admit a Taylor series in $\hbar$:

$$a_i(x;\hbar)=\sum_{k=0}^{\infty}\hbar^k a_{i,k}(x)\,. \tag{82}$$

Again, for the purpose of studying the quantum periods defined by (50), the main problem is to compute $R(x;\hbar)$. We then use the definition (57) of the latter to recast (80) as a a difference equation in Riccati form:

$$\mathcal{R}(x+\hbar;\hbar)\mathcal{R}(x;\hbar)+a_1(x;\hbar)\mathcal{R}(x;\hbar)+a_2(x;\hbar)=0\,. \tag{83}$$

We will henceforth turn our attention to the solution of (83) as a formal series in $\hbar$. The WKB ansatz (49) implies that

$$\mathcal{R}(x;\hbar)=\sum_{k=0}^{\infty}\hbar^k\mathcal{R}_k(x)\,, \tag{84}$$

leading to the following expansion for the difference Riccati equation (83):

$$\sum_{k,l,m=0}^{\infty}\frac{\hbar^{k+l+m}}{l!}\mathcal{R}_k(x)\partial_x^l\mathcal{R}_m(x)+\sum_{k,l=0}^{\infty}\hbar^{k+l}a_{1,k}(x)\mathcal{R}_l(x)+\sum_{k=0}^{\infty}\hbar^k a_{2,k}(x)=0\,. \tag{85}$$

To get this expression we used the expansion

$$\mathcal{R}(x+\hbar;\hbar)=\sum_{k,l=0}^{\infty}\frac{\hbar^{k+l}}{l!}\partial_x^l\mathcal{R}_k(x)\,. \tag{86}$$

An important difference between (85) and the more familiar Riccati equation for second order linear ODEs is that the $\hbar$-expansion (85) contains derivatives of arbitrary order. However, since an $n$-th derivative always comes with a power of $\hbar^n$, the equation at order $n$ contains

---

[13]Although also first-order systems may be presented in terms of higher-order $\hbar$-difference equations by a change of framing [76].

[14]The generalization to higher orders can be pursued along similar lines.

only derivatives of the solution from the previous orders, so that every order can be solved algebraically as in the usual Riccati equation. The solution can be written explicitly:

$$\mathcal{R}_0^{(\pm)}(x) = e^{y_\pm(x)} = -\frac{a_{1,0}}{2} \pm \sqrt{\frac{a_{1,0}^2}{4} - a_{2,0}}\,, \tag{87}$$

corresponding to the two branches of $\Sigma$ in the semiclassical limit (58). Higher orders are entirely fixed by the choice of a branch at level zero

$$\mathcal{R}_n^{(\pm)} = \mp \frac{1}{\sqrt{a_{1,0}^2 - 4a_{2,0}}} \left[ \sum_{m=1}^{n-1} \sum_{l=0}^{m} \frac{1}{l!} \mathcal{R}_{m-l} \partial_x^l \mathcal{R}_{n-m} + \sum_{l=1}^{n} \frac{1}{l!} \mathcal{R}_{n-l} \partial_x^l \mathcal{R}_0 + \sum_{l=1}^{n} a_{1,l} \mathcal{R}_{n-l} + a_{2,n} \right]. \tag{88}$$

While the idea of using $\mathcal{R}(x, \hbar)$ to solve second order order $\hbar$-difference equations is not new, see e.g. [77,78], to our knowledge the systematic WKB solution (88) is absent in the literature. There is no substantial complication in passing from the second order Riccati solution to the higher order one.

## 3.6 Local $\mathbb{P}^1 \times \mathbb{P}^1$

We now use the above formalism to study the $\hbar$-difference equation of local $\mathbb{P}^1 \times \mathbb{P}^1$ and its quantum periods. The quantization of the classical curve (9) is

$$\psi(x + 2\hbar) + \left[ \tau \left( e^{x+\hbar} + e^{-(x+\hbar)} \right) - \kappa \right] \psi(x + \hbar) + \psi(x) = 0\,, \tag{89}$$

which is of general form (80). The associated Riccati equation is therefore

$$\mathcal{R}(x + \hbar; \hbar) \mathcal{R}(x; \hbar) + \left[ \tau \left( e^x + e^{-x} \right) - \kappa \right] \mathcal{R}(x; \hbar) + 1 = 0\,. \tag{90}$$

The first few terms of the general solution (87)-(88) are

$$\begin{aligned}
\mathcal{R}_0^{(\pm)}(x) &= \frac{\kappa}{2} \pm \sqrt{\frac{1}{4}(\kappa - 2\tau \mathrm{Ch}(x))^2 - 1 - \tau \mathrm{Ch}(x)}\,, \\
\mathcal{R}_1^{(\pm)}(x) &= \frac{\tau \mathrm{Sh}(x)}{2\left(\frac{1}{4}(\kappa - 2\tau \mathrm{Ch}(x))^2 - 1\right)}\,, \\
\mathcal{R}_2^{(\pm)}(x) &= \pm \frac{1}{32\left(\frac{1}{4}(\kappa - 2\tau \mathrm{Ch}(x))^2 - 1\right)^{5/2}} \\
&\quad \times \Bigg[ -\tau^4 \mathrm{Ch}(4x) + \kappa \tau^3 \mathrm{Ch}(3x) + \tau^2 \left(\kappa^2 + 4\tau^2 - 6\right) \mathrm{Ch}(2x) \\
&\quad - \kappa \tau \left(\kappa^2 + 13\tau^2 - 4\right) \mathrm{Ch}(x) + \tau^2 \left(5\kappa^2 + 5\tau^2 - 2\right) \Bigg]\,,
\end{aligned} \tag{91}$$

with $\tau$ defined as in (23). Higher order terms quickly increase in complexity. Overall the WKB differential has the following expansion:

$$
\begin{aligned}
\frac{1}{\hbar}\log\mathcal{R}^{(\pm)}(x;\hbar) = {} & \frac{1}{\hbar}\log\left(\frac{\kappa}{2}\pm\sqrt{\frac{1}{4}(\kappa-2\tau\mathrm{Ch}(x))^2-1}-\tau\mathrm{Ch}(x)\right) \\
& + \frac{t\,\mathrm{Sh}(x)}{2\left(\frac{1}{4}(\kappa-2\tau\mathrm{Ch}(x))^2-1\right)\left(\frac{\kappa}{2}\pm\sqrt{\frac{1}{4}(\kappa-2\tau\mathrm{Ch}(x))^2-1}-\tau\mathrm{Ch}(x)\right)} \\
& \pm\hbar\Big[-\tau^4\mathrm{Ch}(4x)+\kappa\tau^3\mathrm{Ch}(3x)+\tau^2\left(5\kappa^2+5\tau^2-2\right)+\tau^2\left(\kappa^2+4\tau^2-6\right)\mathrm{Ch}(2x) \\
& -\kappa\tau\left(\kappa^2+13\tau^2-4\right)\mathrm{Ch}(x)\Big]\frac{1}{32}\left(\frac{1}{4}(\kappa-2\tau\mathrm{Ch}(x))^2-1\right)^{-5/2} \\
& \times\left(\frac{\kappa}{2}\pm\sqrt{\frac{1}{4}(\kappa-2\tau\mathrm{Ch}(x))^2-1}-\tau\mathrm{Ch}(x)\right)^{-1} \\
& -\hbar\frac{\tau^2\mathrm{Sh}^2(x)}{8\left(\frac{1}{4}(\kappa-2\tau\mathrm{Ch}(x))^2-1\right)^2\left(\frac{\kappa}{2}\pm\sqrt{\frac{1}{4}(\kappa-2\tau\mathrm{Ch}(x))^2-1}-\tau\mathrm{Ch}(x)\right)^2}+O\left(\hbar^2\right).
\end{aligned}
\tag{92}
$$

This solution for the quantum differential allows to compute the wavefunction on the one hand, and quantum periods on the other hand.[15] We will focus on the computation of quantum periods, which involves integrating $\log\mathcal{R}$ along closed cycles as in (62). To this end, a few remarks are in order:

- The only term with logarithmic branching is the classical one, while the other terms have only square root cuts, so we do not have to worry about the logarithmic branching when talking about the quantum corrections.

- One may define, in analogy with the case of second order ODEs, the even and odd differentials under the hyperelliptic involution exchanging the two sheets of the square root

$$
S_{\mathrm{odd}}:=\frac{1}{2}\left(\log\mathcal{R}^{(+)}-\log\mathcal{R}^{(-)}\right),\qquad S_{\mathrm{even}}:=\frac{1}{2}\left(\log\mathcal{R}^{(+)}+\log\mathcal{R}^{(-)}\right).
\tag{93}
$$

  In the differential case, the even contribution is a total derivative, a fact that can be proven using Riccati equation [20]. Even though no such proof is available to our knowledge in the $\hbar$-difference case, it seems to be also true in this example that $S_{even}$ is a total derivative. We checked this statement up to order $O(\hbar^9)$. It is worth noting that this could be stemming from the fact the classical curve (9) is invariant under $y\to-y$, so that $e^{y^{(+)}(x)+y^{(-)}(x)}=1$, so that the classical differential does not have an even component.

- By direct inspection, it also appears that even powers in the $\hbar$-expansion of $S_{odd}$ are total derivatives. For the purpose of computing quantum periods, the WKB differential can then be taken to be just the odd $\hbar$-expansion of $S_{odd}$, in complete analogy with the differential case. It would be interesting to understand this fact from a more general point of view.

Despite these simplifications, direct computation of expressions for the WKB differentials become quickly very unwieldy. A way around this problem is the so-called "quantum operator method", first discovered in [79] for the case of the Modified Mathieu equation. It consists in

---

[15]The computation of a wavefunction involves the choice of suitable boundary condition, as discussed in Section 3.3. In this example it is clear that $\psi_0(x)$ and $\psi_\infty(x)$ will not be constant along either branch of the classical curve. Since we are mainly interested in quantum periods, we will not discuss solutions for $\psi$ further.

writing the higher order periods as linear combinations of the classical ones and their derivatives with respect to the moduli of the curve, and was used in [23] to compute the first corrections to the classical periods in the WKB expansion of local $\mathbb{P}^1 \times \mathbb{P}^1$. While this method can be pushed to very high orders in the context of ordinary WKB expansion [80,81], it seems to be too computationally expensive in the $\hbar$-difference case, an issue that will be circumvented in Section 5 by the use of q-Painlevé equations.

**Exact limits**

To conclude our analysis of the WKB quantum periods on a higher note, we remark that there exist certain limits of the geometry in which an exact computation becomes feasible.

1.  The first one is the region characterized by $\kappa \to 0$ with $\tau$ finite. In this limit it can be seen by explicit computation (although we do not have a proof of this statement to all orders) that the higher-order corrections become total derivatives. This can be checked explicitly by taking the limit $\kappa \to 0$ in (92). The first resulting order of the even and odd differentials are

$$
\begin{aligned}
S_{even} &= \left( -\frac{\tau^2 \mathrm{Sh}(2x)}{\tau^2 + \tau^2 \mathrm{Ch}(2x) - 2} - \hbar \frac{\tau^2 \left( \tau^2 + \left( \tau^2 - 2 \right) \mathrm{Ch}(2x) \right)}{\left( \tau^2 + \tau^2 \mathrm{Ch}(2x) - 2 \right)^2} \right) dx + O(\hbar^2) \\
&= -\frac{1}{2} \mathrm{d} \log \left( \tau^2 + \tau^2 \mathrm{Ch}(2x) - 2 \right) \\
&\quad - \hbar \, \mathrm{d} \left( \frac{\tau^2 \mathrm{Sh}(2x)}{2 \left( \tau^2 + \tau^2 \mathrm{Ch}(2x) - 2 \right)} \right) + O(\hbar^2),
\end{aligned}
\tag{94}
$$

$$
\begin{aligned}
S_{odd} &= \frac{2\pi R}{\hbar} \left( \lambda^{(+)} - \lambda^{(-)} \right) - \frac{\sqrt{2} \tau \mathrm{Sh}(x)}{\sqrt{\tau^2 + \tau^2 \mathrm{Ch}(2x) - 2}} \\
&\quad - \hbar \frac{\tau^3 \mathrm{Ch}(x) \left( 2\tau^2 + \left( 2\tau^2 - 5 \right) \mathrm{Ch}(2x) + 1 \right)}{\sqrt{2} \left( \tau^2 + \tau^2 \mathrm{Ch}(2x) - 2 \right)^{5/2}} \\
&= \frac{2\pi R}{\hbar} \left( \lambda^{(+)} - \lambda^{(-)} \right) - \mathrm{d} \log \left( \sqrt{2} \tau \mathrm{Ch}(x) + \sqrt{\tau^2 + \tau^2 \mathrm{Ch}(2x) - 2} \right) \\
&\quad - \hbar \, \mathrm{d} \left( \frac{\tau^3 \left( 6\tau^2 - 5 \right) \mathrm{Sh}^3(x) + 6\tau^3 \left( \tau^2 - 1 \right) \mathrm{Sh}(x)}{3\sqrt{2} \left( \tau^2 - 1 \right) \left( \tau^2 + \tau^2 \mathrm{Ch}(2x) - 2 \right)^{3/2}} \right) + O(\hbar^2),
\end{aligned}
\tag{95}
$$

where the total derivatives are single-valued on $\Sigma$, and

$$
\lambda^{(\pm)} = \log \left( \pm \sqrt{\tau^2 \mathrm{Ch}^2(x) - 1} - \tau \mathrm{Ch}(x) \right) \mathrm{d}x \,.
\tag{96}
$$

Since the higher orders are total derivatives, the quantum periods are classically exact:

$$
\lim_{\kappa \to 0} \Pi_\gamma = \frac{2\pi R}{\hbar} \lim_{\kappa \to 0} Z_\gamma \,.
\tag{97}
$$

Note that the limit considered here coincides precisely with the geometric realization of the fine-tuned stratum (23). As we will see in the next section the solutions to TBA (which we conjecture to coincide with WKB quantum periods) are in fact exactly semi-classical for this choice of stability condition. In this light, the observation that higher-order corrections to WKB quantum periods seem to vanish matches exactly with expectations from the proposed identification with TBA solutions.

2. The other asymptotics we will discuss is the limit $\tau \to 0$ describing the degeneration of local $\mathbb{P}^1 \times \mathbb{P}^1$ to the half-geometry (44) ($t \to 0$ limit in (92)). Recall from Section 2.3 the charge lattice of local $\mathbb{P}^1 \times \mathbb{P}^1$ halves in dimension since certain cycles (such as $\gamma_{D_4}$) become infinitely large. In this limit, the $\hbar$-expansion becomes classically exact (note for example that all the higher order corrections to the WKB differential in (92) are proportional to $\tau \to 0$). The periods along the surviving compact cycles are then simply the classical periods of the half-geometry that we computed in (77), (79). Conversely the periods along non-compact cycles (such as $\Pi_{\gamma_{D_4}}$) become divergent integrals over open contours.

# 4 TBA equations

We now turn to the discussion of the Thermodynamic Bethe Ansatz (TBA) equations associated to BPS structures.

## 4.1 Background

The relation between TBA equations and BPS states first appeared in the context of four-dimensional $\mathcal{N} = 2$ theories on $S^1 \times \mathbb{R}^3$. Circle compactifications of 4d $\mathcal{N} = 2$ theories are described by 3d $\mathcal{N} = 4$ sigma models with hyperkähler target $\mathcal{M}$ [82], whose metric receives corrections from 4d BPS particles. A way to encode these corrections is to adopt a twistor description based on a set of Darboux coordinates $Y_{\gamma_i}$ [1]. In turn, these Darboux coordinates are characterized by TBA equations equivalent to a Riemann-Hilbert problem associated to the BPS spectrum. This description of hyperkähler geometry in terms of topological data of BPS spectra was soon realized to be an instance of a more general story. A similar construction was applied to the case of D-instanton corrections to hypermultiplet moduli spaces in type II string theory [83], see [14] for a review. Even more generally, a class of Riemann-Hilbert problems connected to BPS counting was defined in [55, 57, 84], and further studied in [17, 18, 27].

Most relevant to our work is the connection between TBA equations and five-dimensional gauge theories studied in [85–87]. In this context, solutions to TBA equations should be related to quantum periods of $\hbar$-difference equations considered in the previous section. The motivation for this expectation comes from extending an observation of [24] connecting RH problems and ODEs arising in class $S$ theories, to Kaluza-Klein 4d $\mathcal{N} = 2$ theories.[16]

The main goal of this section is to provide evidence that the solutions of TBA equations associated to BPS structures of 5d $\mathcal{N} = 1$ gauge theories coincide, under suitable assumptions, with the quantum periods of the corresponding $\hbar$-difference equations studied in the previous section

$$Y_\gamma = X_\gamma \,, \tag{98}$$

with $X_\gamma$ as defined in (51). Below we will provide supporting evidence for this relation in a few examples. Besides this, there are also heuristic reasons to expect such a relation to hold. One of these is the fact that TBA equations are characterized by certain discontinuities encoded by BPS states, and the same discontinuities are expected to be a feature of quantum periods of $\hbar$DEs. Indeed the latter are asymptotic series in $\hbar$, with leading exponential behavior determined by the classical differential (11). The Stokes graph coincides with the exponential network, whose abelianization map jumps by Stokes-like automorphisms [41, 93]. Another reason is the expectation that the ODE/IM correspondence of [25, 26] should admit an extension to

---

[16]In the 'conformal limit' studied in [19], and for a certain class of ODEs, this relation can be understood as a generalization of the ODE/IM correspondence [25, 26]. The full extent of the relation between TBA equations and ODEs is not fully understood, and is a subject of active investigations [80, 81, 88–92].

$\hbar$DEs, see [94] for a recent discussion. Yet another general motivation is that a similar relation between quantum periods and TBA systems is known to hold for certain 4d $\mathcal{N} = 2$ theories, and 5d $\mathcal{N} = 1$ theories on a circle can be regarded as 4d $\mathcal{N} = 2$ theories of Kaluza-Klein type [40, 56].

## 4.2 Integral equations in the conformal limit

Viewing a 5d $\mathcal{N} = 1$ theory on $S^1 \times \mathbb{R}^4$ as a 4d $\mathcal{N} = 2$ Kaluza-Klein theory, we consider compactifiaction on a further circle of radius $\tilde{R}$, down to $T^2 \times \mathbb{R}^3$. Denoting by $Z_\gamma$ the 4d $\mathcal{N} = 2$ central charge, and by $\theta_\gamma$ the Wilson-'t Hooft lines on the circle $\tilde{S}^1$ taking us from 4d to 3d, we define the 'semiflat' variables following [82]

$$Y_\gamma^{sf}(\zeta) = \exp\left( \frac{\pi \tilde{R}}{\zeta} Z_\gamma + i\theta_\gamma + \pi \tilde{R} \zeta \overline{Z}_\gamma \right). \tag{99}$$

The functions $Y_\gamma(\zeta)$ are then defined by a set of coupled nonlinear integral equations

$$Y_\gamma(\zeta) = Y_\gamma^{sf}(\zeta) \exp\left( -\frac{1}{4\pi i} \sum_{\gamma'} \Omega(\gamma', u)\langle \gamma, \gamma'\rangle \int_{\ell_{\gamma'}} \frac{d\zeta'}{\zeta'} \frac{\zeta' + \zeta}{\zeta' - \zeta} \log(1 - \sigma(\gamma')Y_{\gamma'}(\zeta')) \right), \tag{100}$$

where $\sigma(\gamma) = -1$ if $\Omega(\gamma) = 1$ and $\sigma(\gamma) = 1$ if $\Omega(\gamma) = -2$, while $\ell_\gamma := Z_\gamma \mathbb{R}_-$.

We restrict to the so-called Hitchin section by setting $\theta_\gamma = 0$. An important consequence of this restriction is that $Y_{-\gamma}(-\zeta) = Y_\gamma(\zeta)$. We can then simplify the equations by using CPT symmetry of the BPS spectrum $\Omega(\gamma, u) = \Omega(-\gamma, u)$ to obtain

$$Y_\gamma(\zeta)|_{\theta_\gamma = 0} = Y_\gamma^{sf}(\zeta) \exp\left( -\frac{\zeta}{\pi i} \sum_{\gamma' > 0} \Omega(\gamma', u)\langle \gamma, \gamma'\rangle \int_{\ell_{\gamma'}} \frac{d\zeta'}{(\zeta')^2 - (\zeta)^2} \log(1 - \sigma(\gamma')Y_{\gamma'}(\zeta')) \right), \tag{101}$$

where $\gamma' > 0$ corresponds to the 'positive half' of the charge lattice, defined by $Z_{\gamma'} \in \mathbb{H}$ for some choice of half-plane $\mathbb{H} \subset \mathbb{C}$. To take the conformal limit we replace $\zeta = \epsilon \pi \tilde{R}$ and take $\tilde{R} \to 0$ with $\epsilon$ fixed

$$\log Y_\gamma(\epsilon) = \frac{Z_\gamma}{\epsilon} - \frac{\epsilon}{\pi i} \sum_{\gamma' > 0} \Omega(\gamma', u)\langle \gamma, \gamma'\rangle \int_{\ell_{\gamma'}} \frac{d\epsilon'}{(\epsilon')^2 - (\epsilon)^2} \log(1 - \sigma(\gamma')Y_{\gamma'}(\epsilon')). \tag{102}$$

Varying $\epsilon$ across one of the rays $\ell_{\gamma'}$ induces the solutions $Y_\gamma(\epsilon)$ to jump by a Kontsevich-Soibelman transformation [55]

$$Y_\gamma \to Y_\gamma (1 - \sigma(\gamma')Y_{\gamma'})^{\Omega(\gamma')\langle \gamma, \gamma'\rangle}. \tag{103}$$

## 4.3 The TBA system for local $\mathbb{P}^1 \times \mathbb{P}^1$

Since $Y_\gamma$ obey the product rule

$$Y_\gamma Y_{\gamma'} = Y_{\gamma + \gamma'}, \tag{104}$$

we can always decompose $Y_\gamma = \prod_i Y_{\gamma_i}^{n_i}$ for a choice of generators of the charge lattice $\gamma$ in which $\gamma = \sum_i n_i \gamma_i$. It follows that one only needs to solve TBA equations for $Y_{\gamma_i}$ for $i = 1, \ldots, 4$. Moreover, in the case of local $\mathbb{P}^1 \times \mathbb{P}^1$ the TBA system reduces further, to the computation of two out of four of these variables. Recall from Section 2 that the charge lattice contains a two-dimensional flavour sublattice $\Gamma_f$. The variables $Y_{\gamma_f}$ for $\gamma_f \in \Gamma_f$ do not receive instanton corrections in the TBA equations, since flavour charges have trivial pairing with all charges. We then rotate to a basis (see (16))

$$\gamma_1, \gamma_2, \gamma_{D0}, \gamma_{D2_f \overline{D2_b}}, \tag{105}$$

that reflects the splitting into gauge and flavour charges

$$\Gamma = \Gamma_g \oplus \Gamma_f \,. \tag{106}$$

Then TBA equations for the BPS chamber described in Section 2.2 take the following form

$$Y_{\gamma_{D0}}(\epsilon) = \exp \frac{Z_{\gamma_{D0}}}{\epsilon} \,, \qquad Y_{\gamma_{D2_f \overline{D2}_b}}(\epsilon) = \exp \frac{Z_{\gamma_{D2_f \overline{D2}_b}}}{\epsilon} \,, \tag{107}$$

$$
\begin{aligned}
\log Y_{\gamma_1}(\epsilon) = {} & \frac{Z_{\gamma_1}}{\epsilon} \\
& + \frac{2\epsilon}{\pi i} \sum_{k \geq 0} k \int_{\ell_{\gamma_1 + k(\gamma_1 + \gamma_2)}} \frac{d\epsilon'}{(\epsilon')^2 - (\epsilon)^2} \log\left[1 + Y_{\gamma_1 + k(\gamma_1 + \gamma_2)}(\epsilon')\right] \\
& - \frac{2\epsilon}{\pi i} \sum_{k \geq 0} k \int_{\ell_{\gamma_3 + k(\gamma_3 + \gamma_4)}} \frac{d\epsilon'}{(\epsilon')^2 - (\epsilon)^2} \log\left[1 + Y_{\gamma_3 + k(\gamma_3 + \gamma_4)}(\epsilon')\right] \\
& + \frac{2\epsilon}{\pi i} \sum_{k \geq 0} (k+1) \int_{\ell_{\gamma_2 + k(\gamma_1 + \gamma_2)}} \frac{d\epsilon'}{(\epsilon')^2 - (\epsilon)^2} \log\left[1 + Y_{\gamma_2 + k(\gamma_1 + \gamma_2)}(\epsilon')\right] \\
& - \frac{2\epsilon}{\pi i} \sum_{k \geq 0} (k+1) \int_{\ell_{\gamma_4 + k(\gamma_3 + \gamma_4)}} \frac{d\epsilon'}{(\epsilon')^2 - (\epsilon)^2} \log\left[1 + Y_{\gamma_4 + k(\gamma_3 + \gamma_4)}(\epsilon')\right] \\
& + \frac{2\epsilon}{\pi i} \sum_{k \geq 1} k \int_{\mathbb{R}_{<0}} \frac{d\epsilon'}{(\epsilon')^2 - (\epsilon)^2} \log\left[\frac{1 - Y_{\gamma_1 + \gamma_2 + k\gamma_{D0}}(\epsilon')}{1 - Y_{\gamma_3 + \gamma_4 + k\gamma_{D0}}(\epsilon')}\right] \,,
\end{aligned}
\tag{108}
$$

$$
\begin{aligned}
\log Y_{\gamma_2}(\epsilon) = {} & \frac{Z_{\gamma_2}}{\epsilon} \\
& - \frac{2\epsilon}{\pi i} \sum_{k \geq 0} (k+1) \int_{\ell_{\gamma_1 + k(\gamma_1 + \gamma_2)}} \frac{d\epsilon'}{(\epsilon')^2 - (\epsilon)^2} \log\left[1 + Y_{\gamma_1 + k(\gamma_1 + \gamma_2)}(\epsilon')\right] \\
& + \frac{2\epsilon}{\pi i} \sum_{k \geq 0} (k+1) \int_{\ell_{\gamma_3 + k(\gamma_3 + \gamma_4)}} \frac{d\epsilon'}{(\epsilon')^2 - (\epsilon)^2} \log\left[1 + Y_{\gamma_3 + k(\gamma_3 + \gamma_4)}(\epsilon')\right] \\
& - \frac{2\epsilon}{\pi i} \sum_{k \geq 0} k \int_{\ell_{\gamma_2 + k(\gamma_1 + \gamma_2)}} \frac{d\epsilon'}{(\epsilon')^2 - (\epsilon)^2} \log\left[1 + Y_{\gamma_2 + k(\gamma_1 + \gamma_2)}(\epsilon')\right] \\
& + \frac{2\epsilon}{\pi i} \sum_{k \geq 0} k \int_{\ell_{\gamma_4 + k(\gamma_3 + \gamma_4)}} \frac{d\epsilon'}{(\epsilon')^2 - (\epsilon)^2} \log\left[1 + Y_{\gamma_4 + k(\gamma_3 + \gamma_4)}(\epsilon')\right] \\
& - \frac{2\epsilon}{\pi i} \sum_{k \geq 1} k \int_{\mathbb{R}_{<0}} \frac{d\epsilon'}{(\epsilon')^2 - (\epsilon)^2} \log\left[\frac{1 - Y_{\gamma_1 + \gamma_2 + k\gamma_{D0}}(\epsilon')}{1 - Y_{\gamma_3 + \gamma_4 + k\gamma_{D0}}(\epsilon')}\right] \,,
\end{aligned}
\tag{109}
$$

where we used the spectrum (22) and the intersection pairing (15). Note that the coupled system (108)-(109) can be written entirely in terms of $Y_{\gamma_1}, Y_{\gamma_2}$ by substitution

$$\gamma = \sum_i n_i \gamma_i \quad \Rightarrow \quad Y_\gamma = Y_{\gamma_1}^{n_1 - n_3} Y_{\gamma_2}^{n_2 - n_4} \exp\left\{\frac{1}{\epsilon}\left(n_3 Z_{\gamma_{D0}} + (n_4 - n_3) Z_{\gamma_{D2_f \overline{D2}_b}}\right)\right\} \,. \tag{110}$$

In fact $Y_{\gamma_3}, Y_{\gamma_4}$ can be recovered from solutions of (108)-(109) as follows

$$\log Y_{\gamma_3} = \epsilon^{-1}\left(Z_{\gamma_{D0}} - Z_{\gamma_{D2_f \overline{D2}_b}}\right) - \log Y_{\gamma_1} \,, \qquad \log Y_{\gamma_4} = \epsilon^{-1} Z_{\gamma_{D2_f \overline{D2}_b}} - \log Y_{\gamma_2} \,. \tag{111}$$

## 4.4 An exact solution on the fine-tuned stratum

TBA-type equations like (108)-(109) are generally difficult to solve in closed form. A basic exception is the case of uncoupled BPS structures, characterized by the vanishing of (nearly) all pairings $\langle \gamma, \gamma' \rangle = 0$. Explicit solutions for systems of this kind have been studied for example in [1, 16, 27, 95].

At first sight, solving TBA equations for local $\mathbb{P}^1 \times \mathbb{P}^1$ appears to be a formidable task, since the equations are clearly coupled by the nontrivial pairing matrix (15). However something remarkable happens in the fine-tuned stratum $\mathcal{C}_1$ of the collimation chamber, described by (18). There is an exact $\mathbb{Z}_2$ symmetry acting on the BPS spectrum

$$\gamma_1 \longleftrightarrow \gamma_3, \qquad \gamma_2 \longleftrightarrow \gamma_4, \tag{112}$$

both at the level of central charges (18) and at the level of BPS indices (22). This symmetry is moreover preserved by the conformal limit of the TBA equations (102). In order to see this, consider for example the equation for $Y_{\gamma_1}$

$$\begin{aligned}
\log Y_{\gamma_1}(\epsilon) - \frac{Z_{\gamma_1}}{\epsilon} = &\sum_{k \geq 0} \mathcal{I}_{\gamma_1 + k(\gamma_1 + \gamma_2)} + \sum_{k \geq 0} \mathcal{I}_{\gamma_2 + k(\gamma_1 + \gamma_2)} \\
&+ \sum_{k \geq 0} \mathcal{I}_{\gamma_3 + k(\gamma_3 + \gamma_4)} + \sum_{k \geq 0} \mathcal{I}_{\gamma_4 + k(\gamma_3 + \gamma_4)} \\
&+ \sum_{k \geq 0} \mathcal{I}_{\gamma_1 + \gamma_2 + k\gamma_{D0}} + \sum_{k \geq 0} \mathcal{I}_{\gamma_3 + \gamma_4 + k\gamma_{D0}},
\end{aligned} \tag{113}$$

where we arranged the instanton corrections

$$\mathcal{I}_{\gamma'} = -\frac{\epsilon}{\pi i} \Omega(\gamma', u) \langle \gamma_1, \gamma' \rangle \int_{\ell_{\gamma'}} \frac{d\epsilon'}{(\epsilon')^2 - (\epsilon)^2} \log(1 - \sigma(\gamma') Y_{\gamma'}(\epsilon')), \tag{114}$$

into towers of 'positive' states of the BPS spectrum (22) of charge $\gamma'$ such that $\text{Re}\, Z_{\gamma'} > 0$. Next we claim that

$$\begin{aligned}
\mathcal{I}_{\gamma_1 + k(\gamma_1 + \gamma_2)} + \mathcal{I}_{\gamma_3 + k(\gamma_3 + \gamma_4)} &= 0, \\
\mathcal{I}_{\gamma_2 + k(\gamma_1 + \gamma_2)} + \mathcal{I}_{\gamma_4 + k(\gamma_3 + \gamma_4)} &= 0, \\
\mathcal{I}_{\gamma_1 + \gamma_2 + k\gamma_{D0}} + \mathcal{I}_{\gamma_3 + \gamma_4 + k\gamma_{D0}} &= 0.
\end{aligned} \tag{115}$$

For illustration we prove the first identity. Observe that

$$\langle \gamma_1, \gamma_1 + k(\gamma_1 + \gamma_2) \rangle = k\langle \gamma_1, \gamma_2 \rangle = -k\langle \gamma_1, \gamma_4 \rangle = -\langle \gamma_1, \gamma_3 + k(\gamma_3 + \gamma_4) \rangle, \tag{116}$$

where we made use of $\langle \gamma_1, \gamma_3 \rangle = 0$ and other pairings in (15). Also, observe that

$$\Omega(\gamma_1 + k(\gamma_1 + \gamma_2)) = \Omega(\gamma_3 + k(\gamma_3 + \gamma_4)), \tag{117}$$

from (22). Finally observe that

$$Y_{\gamma_1} = Y_{\gamma_3}, \qquad Y_{\gamma_2} = Y_{\gamma_4}, \tag{118}$$

since each pair solves the same set of equations, up to the relabeling (112). Since this relabeling is a symmetry of the parameters $Z_\gamma, \Omega(\gamma)$ that define TBA equations, it follows that the TBA equations for $Y_{\gamma_1}$ and $Y_{\gamma_3}$ are identical, and this implies the above identity. Taken together, the relations (116), (117) and (118) imply the first line of (115) by direct substitution into (114). The second and third line follow from a similar reasoning.

We conclude that for stability condition (18) the coupled TBA equations of local $\mathbb{P}^1 \times \mathbb{P}^1$ simplify dramatically, and have the exact solution

$$\log Y_{\gamma_i} \equiv \frac{Z_{\gamma_i}}{\epsilon}, \qquad i = 1, \ldots, 4.$$ (119)

Recalling that central charges can be determined exactly in terms of moduli on the fine-tuned stratum, as given in (19) we may write down the quantum periods more explicitly as

$$Y_{\gamma_1} = Y_{\gamma_3} = e^{\frac{\pi}{R\epsilon}} \tau^{-\frac{i}{R\epsilon}}, \qquad Y_{\gamma_2} = Y_{\gamma_4} = \tau^{\frac{i}{R\epsilon}}.$$ (120)

A few comments are in order:

1. It is remarkable that TBA equations for a coupled BPS structure admit such a simple solution. The key feature of this system that makes it possible to simplify TBA equations in this way is the symmetry of the central charges in (18). This is the hallmark of (fine-tuned strata in) collimation chambers defined in [39]. Similar arguments apply to the other geometries studied in our earlier work, namely local Del Pezzo surfaces.

2. We observed at the end of the previous section that all quantum corrections to quantum periods appeared to vanish in the limit $\kappa \to 0$ with $\tau$ fixed, see equation (97). At the same time, recall that the same condition on the complex moduli of the curve appeared in the realization (23) of the fine-tuned stratum (18). Here we have shown that such a configuration of central charges implies directly that solutions to TBA equations are purely semiclassical. Comparing (120) with (97) provides evidence for the proposed identification (98), by showing that each set of functions behaves semiclassically (in the respective parameters $\hbar, \epsilon$) in the same region of the moduli space. We propose to identify

$$\epsilon = \frac{\hbar}{2\pi R} \qquad \longleftrightarrow \qquad \Pi_\gamma \equiv \log X_\gamma = \log Y_\gamma.$$ (121)

3. The exact solution (120) to the TBA equations holds over the whole fine-tuned stratum (18) of the collimation chamber. This corresponds to setting $\delta = 0$ and varying $Z_{\gamma_1}$. As shown in Section 2.3, by taking the limit $Z_{\gamma_1} \to i\infty$ leads to the degeneration into a half-geometry. Therefore (120) also describes solutions to TBA equations arising from BPS structures of $O_{\mathbb{P}^1}(-2) \oplus O_{\mathbb{P}^1}(0)$. Comparing with the quantum periods (77) and (79) obtained in Section 3.4, we again find agreement with the identification (121).

## 4.5 A q-Painlevé appetizer

From the discussion in the previous subsection, the reader may get the impression that TBA equations of local $\mathbb{P}^1 \times \mathbb{P}^1$ become essentially trivial in the collimation chamber. This is not quite true. To arrive at the exact solution (120) we took two crucial steps: i) we worked in the conformal limit, and in particular on the 'Hitchin section' in the sense of [24]; ii) we chose stability conditions from the fine-tuned stratum (18) of the collimation chamber. These two conditions underlie the $\mathbb{Z}_2$ symmetry (112) that led us to the simple exact solution (120). However, relaxing either of these conditions immediately brings back all the complexity that is characteristic of these systems of coupled integral equations.

Let us consider a deformation of the stability condition away from the fine-tuned stratum, but still belonging to the collimation chamber. For a concrete example, see the classes of stability conditions $C_1^{(\delta)}, C_1^{(\rho)}$ discussed in Section 2.2.3. The BPS spectrum (22) is organized

into 'peacock patterns' as in Figure 4. A consequence of this is the existence of an affine $\mathbb{Z}$-symmetry on the BPS spectrum

$$
\begin{aligned}
T(Z_{\gamma_1+n(\gamma_1+\gamma_2)}) &= Z_{\gamma_1+(n+1)(\gamma_1+\gamma_2)}, & T(Z_{\gamma_2+n(\gamma_1+\gamma_2)}) &= Z_{\gamma_1+(n-1)(\gamma_1+\gamma_2)}, \\
T(Z_{\gamma_3+n(\gamma_3+\gamma_4)}) &= Z_{\gamma_3+(n+1)(\gamma_3+\gamma_4)}, & T(Z_{\gamma_4+n(\gamma_3+\gamma_4)}) &= Z_{\gamma_3+(n-1)(\gamma_3+\gamma_4)}.
\end{aligned}
\tag{122}
$$

Under this action, both the black and the red tower of states in the upper part of Figure 4 simply shift to the right by one unit. Towers at the bottom shift left by one unit. The states in the middle stay put. The BPS spectrum is therefore invariant, in the sense of (33), under the action of $T$ on central charges followed by a relabeling of $\gamma$'s defined in (32).

This affine symmetry of the BPS spectrum leads to an interesting constraint for the solutions of TBA equations. Indeed, in the conformal limit (and on the Hitchin section) the latter are entirely determined by $\Omega(\gamma)$ and $Z_\gamma$, the same data that defines the BPS spectrum. For convenience let us denote a choice of stability condition in $\tilde{\mathcal{C}}_1$ by $Z_\gamma$ and let us denote by $\overline{Z}_\gamma$ the image under $T$, while the image under $T^{-1}$ will be denoted by $\underline{Z}_\gamma$. We may rewrite (122) as

$$
\overline{Z}_{\gamma_1} = Z_{\gamma_1+(\gamma_1+\gamma_2)}, \qquad \overline{Z}_{\gamma_2} = Z_{\gamma_2-(\gamma_1+\gamma_2)},
\tag{123}
$$

and similarly for central charges involving $\gamma_3, \gamma_4$.

The two stability conditions defined by $Z_\gamma$ and $\overline{Z}_\gamma$ are connected by the continuous path

$$
Z_\gamma(s) = (1-s)Z_\gamma + s\overline{Z}_\gamma, \qquad 0 \le s \le 1.
\tag{124}
$$

Along this path, the slopes of BPS rays $\ell_\gamma$ that define integration contours for TBA equations (109)-(109) rotate. There are critical moments $0 < s(\gamma_1), s(\gamma_3) < 1$ for which the rays $\ell_{\gamma_1}, \ell_{\gamma_3}$ cross the phase of $\epsilon$. At these moments, the values of $Y_{\gamma'}$ with nonzero pairing with $\gamma_1$ or $\gamma_3$ jump. Recalling the pairing matrix (15) we obtain[17]

$$
\overline{Y}_{\gamma_1} = Y_{\gamma_1+(\gamma_1+\gamma_2)} \cdot \left(\frac{1+Y_{\gamma_3}}{1+Y_{\gamma_1}}\right)^2,
\tag{125}
$$

$$
\overline{Y}_{\gamma_2} = Y_{-\gamma_1}.
$$

The affine $\mathbb{Z}$-symmetry allows to deduce from the second equation that $Y_{\gamma_2} = \underline{Y}_{-\gamma_1}$ holds as well. Then using the basic properties $Y_\gamma Y_{\gamma'} = Y_{\gamma+\gamma'}$ and $Y_{-\gamma} = Y_\gamma^{-1}$, and combining with the first equation gives

$$
\overline{Y}_{\gamma_1}\underline{Y}_{-\gamma_1} = \left(\frac{Y_{\gamma_1}+Y_{\gamma_1+\gamma_3}}{1+Y_{\gamma_1}}\right)^2.
\tag{126}
$$

Finally, recall from (16) that $\gamma_1 + \gamma_3$ is a pure-flavor charge. The TBA solution for $Y_{\gamma_1+\gamma_3}$ can be written exactly in terms of (107) as

$$
Y_{\gamma_1+\gamma_3} = \exp\left(\frac{1}{\epsilon}(Z_{\gamma_{D0}} - Z_{\gamma_{D2_f}\overline{D2_b}})\right) = \exp\left(\frac{2\pi}{\epsilon R} - \frac{2i}{\epsilon R}\log\tau\right).
\tag{127}
$$

Using this we may rewrite the equation entirely in terms of $Y_{\gamma_1}$

$$
\overline{Y}_{\gamma_1}\underline{Y}_{-\gamma_1} = \left(\frac{Y_{\gamma_1}+e^{\frac{2\pi}{\epsilon R}}\tau^{-\frac{2i}{\epsilon R}}}{Y_{\gamma_1}+1}\right)^2.
\tag{128}
$$

---

[17]Let us briefly comment on how this equation is derived. The TBA equations for stability conditions $Z_\gamma$ and $T(Z_\gamma)$ are nearly identical. They involve the same BPS spectrum, thanks to the symmetry (33), therefore the coefficients in the equations are the same. Naively this would lead to $\overline{Y}_{\gamma_1} = Y_{\gamma_1+(\gamma_1+\gamma_2)}$. However there is one difference between the equations before and after the $T$-action, which accounts for the correction in (125). Since $\epsilon$ is kept fixed, the BPS rays of $\ell_{\gamma_1}$ and $\overline{\ell}_{\gamma_1} \equiv \ell_{\gamma_1+(\gamma_1+\gamma_2)}$ lie on opposite sides of the $\mathbb{H}_\epsilon$ half-plane boundary. This induces a shift $Y_{\gamma_2} \to Y_{\gamma_2}\left(\frac{1+Y_{\gamma_3}}{1+Y_{\gamma_1}}\right)^2$ on the right hand side.

This has the form of the q-Painlevé III$_3$ equation associated with the geometry of local $\mathbb{P}^1 \times \mathbb{P}^1$, that we will see in more generality in the next section. Note that (120) is a particular solution, we will return to this point in section 5.3.

We have therefore shown how the q-Painlevé equation can be derived by studying the TBA equations defined by the BPS spectrum of the collimation chamber. A key property of the BPS spectrum that enters this derivation is the $\mathbb{Z}$-symmetry defined by a combination of $T$ and a suitable relabeling of charges. In the setting of q-Painlevé, this operation turns out to be related to a discrete time evolution. There is also a geometric interpretation of this evolution in terms of an affine translation symmetry within the Cremona group of local $\mathbb{P}^1 \times \mathbb{P}^1$, this played an important role in the study of BPS spectra and the definition of collimation chambers [39].

# 5 q-Painlevé cluster coordinates

The connection between q-Painlevé equations and the spectrum of BPS states of local $\mathbb{P}^1 \times \mathbb{P}^1$ was observed in [52] and further studied in [39]. In this section we propose a relation between the quantum periods (50), the solutions $Y_\gamma$ to the TBA equations (102), and solutions to q-Painlevé equations, that allows us to write the quantum periods as a (convergent) series in the Kähler parameters of local $\mathbb{P}^1 \times \mathbb{P}^1$, exact in $\hbar$.

## 5.1 q-Painlevé equations and cluster integrable systems

There is a natural cluster algebra [96,97] associated to the BPS quiver, with adjacency matrix $B_{ij}$ specified by the intersection paring, e.g. (15) for the case of local $\mathbb{P}^1 \times \mathbb{P}^1$. In the case of purely four-dimensional theories, this connection to cluster algebras has been used to determine the BPS spectrum in appropriate chambers through the so-called mutation method [98]. Recall that a quiver is associated both to a choice of stability condition and of a half-plane of central charges, determining a labelling of the quiver nodes by basis elements of the charge lattice. When we vary the moduli it can happen that a central charge $Z_{\gamma_k}$ of the state $\gamma_k \in \Gamma$ exits from our choice of half-plane, while its antiparticle enters from the other side. Alternatively, this can happen if we rotate the choice of half-plane, leaving the moduli unchanged, an operation called *tilting* of the half-plane. We then have a change of charge labels for our quiver nodes, described by a *(left) mutation* of the quiver at the node $k$:

$$\mu_k(\gamma_j) = \begin{cases} -\gamma_j, & j = k, \\ \gamma_j + [B_{kj}]_+ \gamma_k, & \text{otherwise,} \end{cases} \qquad [B_{kj}]_+ := \max\left(B_{kj}, 0\right). \tag{129}$$

$$\mu_k(B_{ij}) = \begin{cases} -B_{ij}, & i = k, \text{ or } j = k, \\ B_{ij} + \frac{B_{ik}|B_{kj}| + B_{kj}|B_{ik}|}{2}, & \text{otherwise.} \end{cases} \tag{130}$$

When there is a finite number of BPS states, it is possible to uncover the whole BPS spectrum by performing a full tilting of the positive half-plane, since every state in the spectrum appears as a node in the quiver at some point. This happens when there are only hypermultiplet states. Any state of spin at least one-half[18] is an accumulation ray for an infinite tower of hypermultiplet states (recall that only spin-0 states appear as nodes of the quiver). However, with a bit of care it is possible to generalize the mutation method to cover the cases with one limiting ray of spin 1/2 as well [99].

---

[18]By 'spin' we refer to the representation of the Clifford vacuum $\mathfrak{h}$ of a short multiplet, under the little group $SO(3)$ in 4d. The BPS multiplet is obtained by tensoring with a universal half-hypermultiplet $\mathfrak{h} \otimes \rho_{hh}$. Hypermultiplets have $\mathfrak{h}$ a spin-0 singlet, vectormultiplets are spin 1/2 doublets, and so on.

In addition to considering the charge vectors $\gamma_i \in \Gamma$, we will introduce the so-called $\mathcal{X}$-cluster variables: if $\{\gamma_i\}$ is a basis for $\Gamma$ and $\gamma := \sum n_i \gamma_i \in \Gamma$ for some integers $n_i$, then we define $\mathcal{X}_\gamma$ as

$$\mathcal{X}_\gamma = \prod_i \mathcal{X}_i^{n_i} \in \mathbb{C}^\times, \qquad \mathcal{X}_{\gamma_i} := \mathcal{X}_i. \tag{131}$$

The adjacency matrix of the quiver defines a Poisson bracket on the so-called $\mathcal{X}$-cluster variety [100], of which the $\mathcal{X}_i$'s are local coordinates:

$$\{\mathcal{X}_i, \mathcal{X}_j\} = B_{ij} \mathcal{X}_i \mathcal{X}_j, \tag{132}$$

while mutations and permutations take value in the algebra of Poisson maps of the discrete integrable system. The mutations on the $\mathcal{X}$-cluster variables are birational transformations defined by

$$\mu_k(\mathcal{X}_j) = \begin{cases} \mathcal{X}_j^{-1}, & j = k, \\ \mathcal{X}_j \left(1 + \mathcal{X}_k^{\mathrm{sgn} B_{jk}}\right)^{B_{jk}}, & \text{otherwise.} \end{cases} \tag{133}$$

The kernel of the adjacency matrix of the quiver is giving the Casimirs of the Poisson algebra, one among which plays a distinguished role:

$$q := \prod_i \mathcal{X}_i. \tag{134}$$

By the map (131) and the identification (14), it is associated to the D0-brane charge vector. From the data of a quiver together with its $\mathcal{X}$-cluster variables, it is possible to introduce a discrete (nonautonomous, cluster) integrable system, whose deautonomization yields q-Painlevé equations [5, 101, 102], with dynamical variables the $\mathcal{X}_i$'s. The discrete time evolution of the system is given by an appropriate sequence of mutations and permutations [5]. Note that discontinuities (103) of solutions of TBA equations when $\Omega = 1$ match exactly with mutations of cluster variables $\mathcal{X}$ (147), that fully determine their time dependence.[19]

It follows then that solutions of the TBA equations must also satisfy the $q$-difference equations of the cluster integrable system, i.e. we identify

$$\mathcal{X}_i \equiv Y_{\gamma_i} \overset{(98)}{=} X_{\gamma_i}. \tag{135}$$

Since the collimation chamber is a highly fine-tuned chamber in moduli space, the corresponding solution to the cluster integrable system will turn out to be of a very special kind. The rest of the section will be devoted to making this statement precise in our example of local $\mathbb{P}^1 \times \mathbb{P}^1$.

## 5.2 Local $\mathbb{P}^1 \times \mathbb{P}^1$

The pairing (15) gives rise to the quiver in Figure 7, where to the $i$-th node is associated the charge $\gamma_i$ of the basis (14). The adjacency matrix of the quiver is the intersection pairing (15):

$$B_{ij} := \langle \gamma_i, \gamma_j \rangle = \begin{pmatrix} 0 & -2 & 0 & 2 \\ 2 & 0 & -2 & 0 \\ 0 & 2 & 0 & -2 \\ -2 & 0 & 2 & 0 \end{pmatrix}. \tag{136}$$

---

[19]Analogous relations between 4d $\mathcal{N} = 2$ gauge theories and discrete integrable systems were investigated in [103, 104].

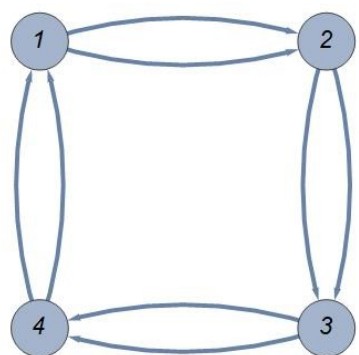

Figure 7: BPS quiver for local $\mathbb{P}^1 \times \mathbb{P}^1$.

The spectrum (22) of local $\mathbb{P}^1 \times \mathbb{P}^1$ in the collimation chamber is produced by acting on the BPS charges with the affine translation on the $A_1^{(1)}$ root lattice [52], realized as

$$T = (1, 2)(3, 4)\mu_1 \mu_3 \,. \tag{137}$$

Indeed, it was shown in [39] that this discrete flow has the interpretation of a tilting of the positive half-plane in the collimation chamber described by (18).

In this case there are two Casimirs

$$t := \mathcal{X}_2^{-1} \mathcal{X}_4^{-1} \,, \qquad q := \mathcal{X}_1 \mathcal{X}_2 \mathcal{X}_3 \mathcal{X}_4 \,. \tag{138}$$

Under the identifications (14), (131), the variable $t$ is associated to $D2_b \overline{D2}_f$, while $q$ is associated to the $D0$ brane.

$$q = \mathcal{X}_1 \dots \mathcal{X}_4 = Y_{D0} = X_{D0} \,, \qquad t = \mathcal{X}_2^{-1} \mathcal{X}_4^{-1} = Y_{D2_b \overline{D2}_f} = X_{D2_b \overline{D2}_f} \,. \tag{139}$$

Since the Casimirs are associated to pure flavour charges, their expression in terms of central charges is purely semiclassical. The first equation above allows us to relate the different deformation parameters, namely the parameter $q$ appearing in q-Painlevé, the parameter $\hbar$ appearing in the WKB expansion, and the parameter $\epsilon$ in the TBA equations:

$$q = \exp\left\{\frac{1}{\epsilon} Z_{\gamma_1 + \gamma_2 + \gamma_3 + \gamma_4}\right\} = e^{\frac{4\pi^2}{\hbar}} = e^{\frac{2\pi}{R\epsilon}} \,. \tag{140}$$

The match between $\hbar$ and $\epsilon$ is the same as (121), found in the discussion of TBA equations. The other Casimir is parametrized as

$$t = \exp\left\{-\frac{1}{\epsilon} Z_{\gamma_2 + \gamma_4}\right\} \,. \tag{141}$$

In the physical slice, where $Z_{\gamma_i}$ are given in terms of the moduli of the curve (9), the relation (138) together with (17) gives

$$t = \tau^{-\frac{2i}{R\epsilon}} = \tau^{-\frac{4\pi i}{\hbar}} \,. \tag{142}$$

Note that, upon sending $\hbar \to -i\hbar$, the identifications (140),(142) coincide with those appearing in the TS/ST correspondence [4]. The action of the time evolution (137) on the Casimirs is[20]

$$T(t) = qt \,, \qquad T(q) = q \,, \tag{143}$$

---

[20]Note that the Casimir $t$ is not preserved by the time evolution. This is because q-Painlevé equations are a *nonautonomous* discrete integrable system, and the discrete time steps define a foliation of the Casimir level surfaces.

corresponding to the motion in the moduli space of stability conditions

$$Z_{\gamma_2+\gamma_4} \to Z_{\gamma_2+\gamma_4} - Z_{D0}, \tag{144}$$

extending the $\mathbb{Z}$-action on the fine-tuned stratum defined in (30), (32) to the whole collimation chamber. On the physical slice, this reduces to

$$\tau \to e^{\pi i}\tau. \tag{145}$$

The time evolution on the $\mathcal{X}$-cluster variables and BPS charges reads

$$\begin{cases} \mathcal{X}_1(qt) = \mathcal{X}_2(t)\left(\frac{1+\mathcal{X}_3(t)}{1+\mathcal{X}_1(t)^{-1}}\right)^2, \\ \mathcal{X}_2(qt) = \mathcal{X}_1(t)^{-1}, \\ \mathcal{X}_3(qt) = \mathcal{X}_4(t)\left(\frac{1+\mathcal{X}_1(t)}{1+\mathcal{X}_3(t)^{-1}}\right)^2, \\ \mathcal{X}_4(qt) = \mathcal{X}_3(t)^{-1}, \end{cases} \begin{cases} T^n(\gamma_1) = \gamma_1 + n(\gamma_1+\gamma_2), \\ T^n(\gamma_2) = \gamma_2 - n(\gamma_1+\gamma_2), \\ T^n(\gamma_3) = \gamma_3 + n(\gamma_3+\gamma_4), \\ T^n(\gamma_4) = \gamma_4 - n(\gamma_3+\gamma_4), \end{cases} \tag{146}$$

and the relabeling in the right column coincides precisely with the one from (32). Due to (138) only two cluster variables are independent. We can choose them to be $\mathcal{X}_1, \mathcal{X}_2$, and write the discrete time evolution as the following system of $q$-difference equations, known as the q-Painlevé $III_3$ equation of symmetry type $A_1^{(1)}$,

$$\mathcal{X}_1(qt)\mathcal{X}_1(q^{-1}t) = \left(\frac{\mathcal{X}_1(t) + qt}{\mathcal{X}_1(t) + 1}\right)^2, \qquad \mathcal{X}_2(qt) = \mathcal{X}_1(t)^{-1}. \tag{147}$$

Note that, using the expressions (140) and (142) for the Casimirs $t, q$, this equation agrees exactly with the one obtained from TBA equations (128) upon identification of $Y$ and $\mathcal{X}$: it can be easily checked that the time evolution $T$ defined in Section 4.5 coincides with (137). The realization in terms of $\mathcal{X}$-cluster variables gives us a "dual" interpretation of the discrete time flow.[21] While in terms of the $\gamma_i$ it is natural to view it as a tilting of the positive half-plane, in terms of the $\mathcal{X}_i$ it takes the form of a discrete motion in moduli space. The general solution to (147) has the following form in terms of 5d supersymmetric partition function [5]:

$$\mathcal{X}_1 = (qt)^{1/2}\left(\frac{Z_D(u,s,q,qt)}{s^{\frac{1}{2}}Z_D(q^{\frac{1}{2}}u,s,q,qt)}\right)^2, \qquad \mathcal{X}_2 = t^{-\frac{1}{2}}\left(\frac{s^{\frac{1}{2}}Z_D(q^{\frac{1}{2}}u,s,q,t)}{Z_D(u,s,q,t)}\right)^2,$$

$$\mathcal{X}_3 = (qt)^{1/2}\left(\frac{s^{\frac{1}{2}}Z_D(q^{\frac{1}{2}}u,s,q,qt)}{Z_D(u,s,q,qt)}\right)^2, \qquad \mathcal{X}_4 = t^{-\frac{1}{2}}\left(\frac{Z_D(u,s,q,t)}{s^{\frac{1}{2}}Z_D(q^{\frac{1}{2}}u,s,q,t)}\right)^2. \tag{148}$$

Here[22]

$$Z_D(u,s,q,t) := \sum_{n\in\mathbb{Z}} s^n Z_{cl}(uq^n,q,t)Z_{pert}(uq^n,q)Z_{inst}(uq^n,q,t), \tag{149}$$

with $Z_{cl}, Z_{pert}, Z_{inst}$ being respectively the tree level, one-loop and instantonic contribution to the 5d gauge theory partition function of pure $SU(2)$ super Yang-Mills [61,105] (here we use the conventions of [52]) with vanishing Chern-Simons level:

$$Z_{cl}(u,q,t) = e^{\log t\left(\frac{\log u}{\log q}\right)^2} = t^{\sigma^2}, \qquad Z_{pert}(u,q) = \left(u^2;q,q^{-1}\right)_\infty\left(u^{-2};q,q^{-1}\right)_\infty, \tag{150}$$

---

[21]More precisely, the tilting is the square of the discrete time step (137): to produce the full spectrum with the tilting one has to use both left and right mutations, while the discrete time evolution appears to be more "fundamental". See [52] for a discussion of this point.

[22]This expression for the tau function is not fully general, since there is a leftover ambiguity from the q-difference equation, see [3, Section 3.2]. However this ambiguity does not affect the variables $\mathcal{X}_i$ therefore we will neglect it in the following.

$$Z_{\text{inst}} = \sum_{\vec{Y}}^{\infty} t^{|\vec{Y}|} Z_{\vec{Y}}, \qquad Z_{\vec{Y}} = \prod_{i,j=1}^{2} \frac{1}{N_{Y_i,Y_j}(u_i/u_j; q, q^{-1})},$$

$$N_{Y,Y'}(u, q_1, q_2) := \prod_{s \in Y} \left(1 - u q_2^{-a_{Y'}(s)-1} q_1^{l_Y(s)}\right) \prod_{s \in Y'} \left(1 - u q_2^{a_Y(s)} q_1^{-l_{Y'}(s)-1}\right). \tag{151}$$

Here we used the notation $u := u_1 = u_2^{-1}$, $\sigma := \log u / \log q$, and $\vec{Y} := (Y_1, Y_2)$ is a pair of partitions, with $a_Y(s)$, $l_Y(s)$ being respectively the arm and leg length of the box $s$ with respect to the partition $Y$. From the gauge theory point of view, $u$ is the electric variable, while $s$ is its dual magnetic variable. From the point of view of q-Painlevé dynamics, they parametrize the choice of integration constants, and can be written as (assuming $\text{Re}\,\sigma > 0$ for simplicity, the other case is analogous):

$$u^2 = \lim_{t \to 0} \mathcal{X}_1(t) \mathcal{X}_2(t), \qquad s = \lim_{t \to 0} \mathcal{X}_1(t)(qt)^{-2\sigma} \left(\frac{Z_{pert}(uq^{-\frac{1}{2}})}{Z_{pert}(u)}\right)^2, \tag{152}$$

as can directly be verified from the $t \to 0$ limit of (148). Together with equation (138), this specifies all parameters appearing in the dual partition function expression (148) in terms of the $\mathcal{X}$-cluster variables and their asymptotics only. By the identification (14), $u, s$ are associated respectively to the $D2_f$ and a regularization of the $D4$ quantum period in the half-geometry limit. This is consistent with the expectation from the expression in terms of Nekrasov-Okounkov dual partition functions, where $u$ is the exponentiated Coulomb modulus, while $s$ is related to its magnetic dual coordinate.

## 5.3 TBA solutions in the collimation chamber via q-Painlevé

We are now ready to find out what is the q-Painlevé solution describing the quantum periods in the collimation chamber. Having already seen that the $Y_\gamma$ satisfy equation (147), we need to identify the correct values for the integration constants $u, s$.

Let us start from the fine-tuned stratum of the collimation chamber, where everything is parametrized in terms of the moduli of the curve (9). For concreteness, let us assume $\text{Im}\,\hbar < 0$, so that $t \to 0$ corresponds to the $\tau \to 0$ half-geometry limit.[23] In fact, $Z_{\gamma_1+\gamma_2}$ is constant in the fine-tuned stratum, so

$$u^2 = \lim_{t \to 0} Y_{\gamma_1} Y_{\gamma_2} = \exp\left\{\frac{2\pi R}{\hbar} \lim_{t \to 0}\left(Z_{\gamma_1} + Z_{\gamma_2}\right)\right\} \overset{(42)}{=} e^{\frac{2\pi^2}{\hbar}} = q^{1/2}. \tag{153}$$

To find what is the value of $s$ corresponding to the Hitchin section over the fine-tuned stratum, we use (120), together with (140), (142), which say $Y_{\gamma_1} = (qt)^{\frac{1}{2}}$ in this case:

$$s = \lim_{t \to 0} Y_{\gamma_1}(qt)^{-2\sigma}\left(\frac{Z_{pert}(uq^{-\frac{1}{2}})}{Z_{pert}(u)}\right)^2 = \left(\frac{Z_{pert}(q^{-\frac{1}{4}})}{Z_{pert}(q^{\frac{1}{4}})}\right)^2 = 1, \tag{154}$$

where we also used the property $Z_{pert}(u) = Z_{pert}(u^{-1})$, following from its definition (150). This is equivalent to $\sigma = \log u / \log q = \frac{1}{4}$. The constraint $u = q^{\frac{1}{4}}$ stems for the restriction $Z_{D2_f} = \frac{\pi}{R}$, which is valid in the fine-tuned stratum. Comparing with the expression (42) for the central charges in the half-geometry, we see that

$$u^2 = q^{1/2} \quad \longleftrightarrow \quad \kappa \to 0. \tag{155}$$

---

[23]Due to (142) the $t \to 0$ asymptotics of the q-Painlevé solution corresponds to either $\tau \to \infty$ or $\tau \to 0$, depending on the sign of the imaginary part of $\hbar$.

These special values $u = q^{1/4}$, $s = 1$ correspond to the so-called *algebraic solution* to q-Painlevé III$_3$ (148) [3, 4], which coincides with the solution to the TBA equations in the fine-tuned stratum:

$$Y_{\gamma_1}\Big|_{u=q^{\frac{1}{4}},s=1} = Y_{\gamma_3}\Big|_{u=q^{\frac{1}{4}},s=1} = (qt)^{\frac{1}{2}} = e^{\frac{2\pi^2}{\hbar}} \tau^{-\frac{2\pi i}{\hbar}}, \tag{156}$$

$$Y_{\gamma_2}\Big|_{u=q^{\frac{1}{4}},s=1} = Y_{\gamma_4}\Big|_{u=q^{\frac{1}{4}},s=1} = t^{-\frac{1}{2}} = \tau^{\frac{2\pi i}{\hbar}}. \tag{157}$$

Since in the fine-tuned locus there are no quantum corrections, from the above cluster variables we can directly read off the central charges by taking a logarithm. What we find reproduces precisely the stability condition (18), thus providing a strong check for the correspondence between q-Painlevé and solutions to the TBA equations.

Let us see what happens when we deform away from the fine-tuned stratum, staying in the collimation chamber. The limit $t \to 0$ in terms of stability data corresponds to sending all the central charges to infinity, while keeping $Z_{\gamma_1+\gamma_2} = Z_{\gamma_{D2_f}}$ and $Z_{D0}$ finite. We will take this limit staying within the same family of stability conditions, so that the spectrum will not jump and will still be described by the q-Painlevé equation as discussed in Section 4.5. Consider first the family of stability conditions $\mathcal{C}_1^{(\delta)}$ in (34), where $Z_{\gamma_3} = Z_{\gamma_1} + \delta$, $Z_{\gamma_1+\gamma_2} = \frac{\pi}{R}$:

$$u^2 = \lim_{t \to 0} Y_{\gamma_1} Y_{\gamma_2} = e^{\frac{2\pi^2}{\hbar}(1+O(\hbar))}. \tag{158}$$

Note that in general $u$ may receive $\hbar$-corrections, since the generic stability condition $\mathcal{C}_1^{(\delta)}$ might not belong to the physical slice of the collimation chamber. The initial condition $s$ is more complicated. To understand how it is related to stability data, we perform an asymptotic expansion in $\epsilon$ *after* the limit $t \to 0$ defining $s$:

$$\begin{aligned} s &= \lim_{t \to 0} Y_{\gamma_1} (qt)^{-2\sigma} \left( \frac{Z_{pert}(uq^{-\frac{1}{2}})}{Z_{pert}(u)} \right)^2 \\ &= \lim_{t \to 0} \exp\left\{ \frac{Z_1}{\epsilon} - \frac{R}{2\pi\epsilon}(Z_1 + Z_3)(Z_1 + Z_2) + O(\epsilon^0) \right\} \times (\text{nonpert. corrections in } \epsilon) \\ &= e^{-\frac{\pi R}{\hbar}\delta(1+O(\hbar))} \times (\text{nonpert. corrections in } \epsilon). \end{aligned} \tag{159}$$

If we consider the family of stability conditions $\mathcal{C}_1^{(\rho)}$, using $Z_3 = \rho Z_1$ and (47), we have

$$u^2 = \lim_{t \to 0} Y_{\gamma_1} Y_{\gamma_2} = e^{\frac{4\pi^2}{\hbar(1+\rho)}(1+O(\hbar))}, \tag{160}$$

while

$$\begin{aligned} s &= \lim_{t \to 0} Y_{\gamma_1} (qt)^{-2\sigma} \left( \frac{Z_{pert}(uq^{-\frac{1}{2}})}{Z_{pert}(u)} \right)^2 \\ &= \lim_{t \to 0} \exp\left\{ \frac{Z_1}{\epsilon} - \frac{R}{2\pi\epsilon}(Z_1 + Z_3)(Z_1 + Z_2) + O(\epsilon^0) \right\} \times (\text{nonpert. corrections in } \epsilon) \\ &= e^{O(\hbar^0)} \times (\text{nonpert. corrections in } \hbar). \end{aligned} \tag{161}$$

Note that $\mathcal{C}_1^{(\delta)}$ correspond to deforming the condition $u = q^{\frac{1}{4}}$ by subleading orders of $\epsilon$, while $s = 1$ is deformed at leading order. The converse happens for $\mathcal{C}_1^{(\rho)}$. Furthermore, since these are both one-parameter deformations of the fine-tuned stratum, in both cases $u, s$ are necessarily not independent. This is likely a consequence of our restriction to the Hitchin section.

Outside the fine-tuned locus, but still within the collimation chamber, we have a highly transcendental solution, as opposed to the algebraic one (156), (157) . For illustration purposes, the first few orders in a small $t$ expansion (corresponding to the half-geometry) read (here we assume that $a = \log u$ has a small positive real part):

$$
\begin{aligned}
\mathcal{X}_2^{\frac{1}{2}} &= \frac{s^{\frac{1}{2}} t^a}{1-u^2} \frac{(u^2;q)_\infty}{(u^{-2};q)_\infty} + \frac{s^{-\frac{1}{2}} t^{-a}}{1-u^{-2}} \frac{(u^{-2};q)_\infty}{(u^2;q)_\infty} \\
&\quad + \left[ \frac{s^{\frac{1}{2}} t^a}{1-u^2} \frac{(u^2;q)_\infty}{(u^{-2};q)_\infty} Z_1(qu^2) + \frac{s^{-\frac{1}{2}} t^{-a}}{1-u^{-2}} \frac{(u^{-2};q)_\infty}{(u^2;q)_\infty} Z_1(q^{-1}u^2) - Z_1(u^2) \right. \\
&\quad \left. + \frac{st^{2a}}{(1-qu^2)(1-u^2)} \frac{(u^2;q)_\infty^2}{(u^{-2};q)_\infty^2} + \frac{s^{-1} t^{-2a}}{(1-qu^{-2})(1-u^{-2})} \frac{(u^{-2};q)_\infty^2}{(u^2;q)_\infty^2} \right] + O(t^2),
\end{aligned}
\tag{162}
$$

where

$$
Z_1(u^2) = \frac{2qu^2}{(1-q)^2(1-u^2)^2} .
\tag{163}
$$

To derive these equations, we expanded the general solution (148) in $t$ and used the following properties:

$$
Z_{inst}(u,q,t) = 1 + t \frac{2}{(1-q^{-1})(1-q)(1-u^{-2})(1-u^2)} + O(t^2),
\tag{164}
$$

$$
\frac{(qz;q,q^{-1})_\infty}{(z;q,q^{-1})_\infty} = (qz;q)_\infty , \qquad\qquad \frac{(qz;q)_\infty}{(z;q)_\infty} = \frac{1}{1-z} .
\tag{165}
$$

**Remark 3.** *There is a manifest difference between the analytic behaviour in $\hbar$ of the TBA solution $Y_\gamma$ and the $q$-dependence of the $q$-Painlevé solution (148). While the $Y_\gamma$'s are piecewise analytic asymptotic series in $\hbar$, the $\mathcal{X}_i$'s appear to be (almost) single-valued functions in $q = e^{\frac{4\pi^2}{\hbar}}$. This is because Nekrasov functions are particular resummations of topological string partition functions [106], and different resummations will be related by Stokes phenomena. The nonperturbative nature of the $q$-Painlevé solutions is already manifest in the $\hbar$-asymptotics of their initial conditions, as in (159), (161).*

### WKB vs TBA/q-Painlevé: A remark on physical stability conditions.

A final remark is in order: we observed in Section 3.6 that for $\kappa \to 0$ with finite $\tau$, the quantum corrections to WKB differentials are exact 1-forms. This leads to semiclassically exact expressions for the quantum WKB periods. A similar conclusion was derived from the viewpoint of TBA equations, by noting that in this limit the classical periods lie on the fine-tuned stratum (see Appendix A), where TBA equations have exact semiclassical solutions (120).

While it is certainly true that the algebraic solution belongs to the physical slice corresponding to the mirror curve (9), at the moment we don't know whether this is true for the solutions corresponding to the families of stability conditions $\mathcal{C}_1^{(\delta)}, \mathcal{C}_1^{(\rho)}$. Nonetheless, it remains true that q-Painlevé cluster variables provide *bona fide* solutions to the TBA equations (102), and Bridgeland-type Riemann-Hilbert problems [57], in the somewhat more general context of the moduli space of stability conditions. This follows from the direct connection between TBA and q-Painlevé derived in Section 4.5. On the other hand, for stability conditions belonging to the physical slice, the limit $t \to 0$ corresponds to a half-geometry. It is then plausible that $u$ and $s$ can be computed exactly, since the BPS structure becomes uncoupled. For $u^2$ we can readily see this: recalling that $\gamma_1 + \gamma_2 = \gamma_{D2_f}$ we still have the exact expression obtained in (77)

$$
u = \lim_{t \to 0} Y_{\gamma_1+\gamma_2} = e^{\Pi_{\gamma_{D2_f}}} = Q^{-\frac{2\pi i}{\hbar}} ,
\tag{166}
$$

where $Q$ is a function of $\kappa$ obtained by inverting (45). From (14) we see that $\gamma_1$ involves the noncompact cycle $\gamma_{D4}$, and its computation requires more care. Possible strategies for its evaluation include the approaches of [27, 72] and [29]. In particular, for the q-Painlevé solution to be physical, the central charges $Z_{D2_f}$, $Z_{D4}$ must be related by special geometry relations.

## 6 Conclusions and Outlook

In this paper we studied a correspondence between TBA equations defined by BPS states of 5d supersymmetric Yang-Mills theory and q-Painlevé equations. We showed that the moduli space of the 5d theory contains a fine-tuned stratum where the BPS spectrum is extremely simple. Using this spectrum to formulate the TBA equations, we explicitly derived the q-Painlevé equations from them. For the families of stability conditions $\mathcal{C}_1^{(\delta)}$, $\mathcal{C}_1^{(\rho)}$ we argued that exact expressions for the TBA solutions $Y_\gamma$ are given in terms of Nekrasov-Okounkov dual partition functions. In the limit $\delta, \rho \to 0$ the stability conditions reduce to that of the fine-tuned stratum $\mathcal{C}_1^{(0)}$, and the TBA solutions simplify significantly, coinciding with the known algebraic solutions of q-Painlevé.

Our work leaves several open questions and raises some directions for future work:

**Cluster coordinates in 4d Kaluza-Klein theories and instanton counting.** The relations between cluster variables $\mathcal{X}_i = Y_{\gamma_i}$ and the gauge theory parameters $(u, s)$ are somewhat reminiscent of the change of variables from Fock-Goncharov to Fenchel-Nielsen that played a role in the identification of tau functions as sections of a certain line bundle over the moduli space of quantum curves in [107] (see in particular equations (5.24)-(5.25) of the reference). This analogy may deserve further study. On the one hand, the theory we study is precisely the KK uplift of 4d $SU(2)$ Yang-Mills studied in [107]. More to the point, the asymptotic limit $t \to 0$ corresponds to a half-geometry where $\tau \to 0$, which is a weak-coupling limit of the 5d gauge theory. As shown in [42, Figure 21], precisely in this limit (and for a suitable choice of phase) the exponential network becomes of 'Fenchel-Nielsen type' in a suitable sense [108–110]. By analogy with the role of spectral coordinates defined by spectral networks [93] in the definition of appropriate decompositions of tau functions in terms of Nekrasov-Okounkov dual partition functions [107, 111], this suggest that a similar role may be played by coordinate systems defined by (non-)Abelianization for exponential networks [41].

**Exact solutions to coupled TBA equations.** It would be interesting to study continuations of the exact solution (120), which holds for moduli (18), to other stability conditions beyond (34), (39). Another possibility would be to rely on differential equations obeyed by $Y_\gamma$ discussed in [1], presumably reinterpreted in connection to the q-Painlevé equations themselves. In the same vein, it would also be interesting to deform away from the Hitchin section by turning on generic $\theta_\gamma \neq 0$. It would also be interesting to see if the explicit solution (148) can be used to compute the Hyperkähler metric on the moduli space, that was the original motivation of [1]. The main complication arises in taking the $\hbar \to 0$ limit of the $\mathcal{X}_i$, which is nontrivial due to the $\hbar$-dependence of $q, t$ in (140), (142). In fact, q-Painlevé solutions contain slightly more information. Even though the q-Painlevé equation itself was obtained from the behavior of the $Y_\gamma$'s, the solution (148) contains nonperturbative corrections that would not appear if we simply solved the TBA order by order in $\hbar$: in this sense, the q-Painlevé solution in terms of Nekrasov-Okounkov dual partition functions is the *resummation* (in $\hbar$) of the TBA solution. This can be seen for example from (162): the expression for $X_2$ contains an infinite number

of nonperturbative terms, with a full series of perturbative contributions attached to each one. In fact, the q-Painlevé initial conditions themselves must depend on $\hbar$ in a nontrivial way determined by the TBA equations. From a resurgence point of view, the presence of subleading trans-series sectors is necessary, to recover the KS formula (103) as the Stokes jumps of the solution, and so must naturally appear in the solution of q-Painlevé (see [80, 81, 110] for a discussion of resurgence and BPS states in four-dimensional pure $SU(2)$ SYM, or e.g. [112] for a general review on resurgent asymptotics).

**Beyond local $\mathbb{P}^1 \times \mathbb{P}^1$.** The analysis carried out in this work relied heavily on explicit knowledge of the BPS spectrum of the 5d gauge theory in the collimation chamber. The same information is available for gauge theories with matter engineered by certain local Del Pezzo surfaces [39]. It would be interesting to extend our analysis to these models. We expect that this should lead to a correspondence between their TBA equations and the solutions of q-Painlevé of "symmetry type" up to $E_5^{(1)}$, corresponding to $dP_5$ [3, 5–7, 52]. In fact, the discussion can also be extended beyond the del Pezzo case, as such expressions are also available for some higher-rank geometries, such as those engineering $SU(N)$ gauge theories [9].

For rank-1 geometries, beyond local $dP_5$ no Nekrasov-Okounkov dual partition functions are available as there is no low-energy gauge theory interpretation. Nonetheless, the q-Painlevé tau functions obey bilinear equations that provide an efficient way to compute the solution order by order in the flavor parameters, that should be related to blowup equations obeyed by the Topological String partition function [8, 113]. Furthermore, taking the four-dimensional limits proposed in [52] on the cluster coordinates would provide exact solutions for the quantum periods also in the purely four-dimensional case.

**Collimation chambers and algebraic solutions** Algebraic solutions of q-Painlevé equations are the invariant solutions with respect to the non-affine part of the corresponding Weyl group (in the case studied in this paper, $W(A_1^{(1)})$): this is the analytic counterpart of the fine-tuning (18) of central charges characterizing these loci in the moduli space, as defined in [39]. It is then quite reasonable to assume that the link between algebraic solutions and fine-tuned collimation chambers can be extended well beyond the case of local $\mathbb{P}^1 \times \mathbb{P}^1$:

**Conjecture 1.** *There exist highly fine-tuned regions in the physical moduli space of five-dimensional theories with underlying affine root lattices where collimation chambers are physically realized, and they are classified by fixed points of the corresponding Weyl group. The quantum periods are semi-classically exact, and given by algebraic solutions of the corresponding cluster integrable system.*

# Acknowledgements

We are grateful to Sibasish Banerjee, Giulio Bonelli, Tom Bridgeland, Cyril Closset, Michele Del Zotto, Harini Desiraju, Pasha Gavrylenko, Alba Grassi, Andy Neitzke, Boris Pioline, Mauricio Romo, Alessandro Tanzini, Jörg Teschner for comments, discussions and correspondence at various stages of this work.

**Funding information** F. D. M. is supported by the Natural Sciences and Engineering Research Council of Canada (NSERC). P. L. is supported by the Knut and Alice Wallenberg Foundation grant KAW2020.0307. Numerical computations were enabled by resources provided by the National Academic Infrastructure for Supercomputing in Sweden (NAISS) and the Swedish

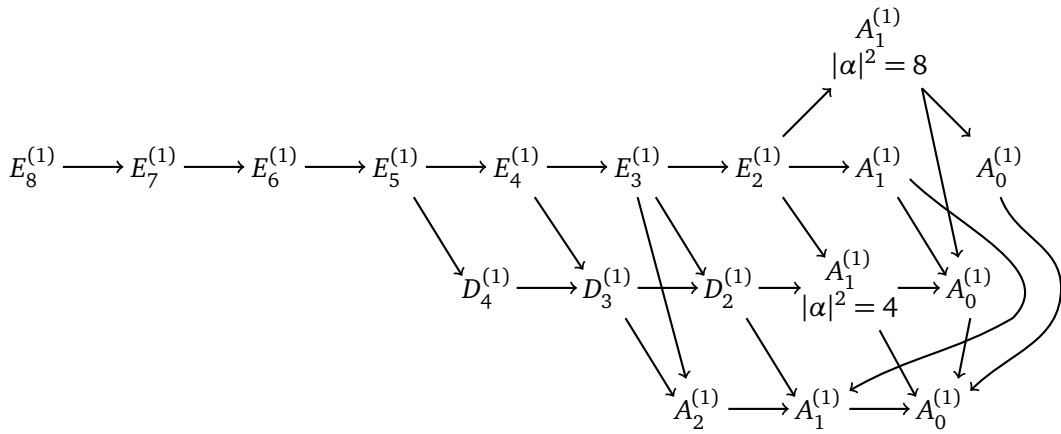

Figure 8: Sakai's Classification of discrete Painlevé equations by symmetry type.

National Infrastructure for Computing (SNIC) at Uppsala University partially funded by the Swedish Research Council through grant agreements no. 2022-06725 and no. 2018-05973.

## A Geometry and BPS states on the fine-tuned stratum

In this Appendix we explore the geometry of the curve (9) in the limit $\kappa \to 0$ with $\tau$ finite. In this limit the curve becomes

$$\tau(e^x + e^{-x}) + e^y + e^{-y} = 0. \tag{A.1}$$

As a double-covering of $\mathbb{C}^*$ with coordinate $e^x$, the curve is two-sheeteted and ramified at four branch points

$$b_{\sigma_1,\sigma_2}: \quad e^x = \frac{\sigma_1 + \sigma_2\sqrt{1-\tau^2}}{\tau}, \tag{A.2}$$

where $\sigma_1, \sigma_2$ are signs $\pm 1$ chosen independently.

Note that, in addition to the obvious symmetries

$$x \to -x, \qquad y \to -y, \tag{A.3}$$

of the generic curve (9), at $\kappa = 0$ there is an additional symmetry generated by

$$(x \to x \pm i\pi, y \to y \pm i\pi), \tag{A.4}$$

with independent choices of signs.

A trivialization of $\Sigma$, including logarithmic branch cuts for $\lambda \sim y\,dx$, is shown in Figure 9. This choice of trivialization is obtained from [42, Figure 4].[24]

The cycle $\gamma_1$ runs between $b_{-,+}$ and $b_{-,-}$ while $\gamma_3$ runs between $b_{+,+}$ and $b_{+,-}$ as shown in Figure 10 (this is for $|\tau| > 1$, an analogous discussion can be carried out for $|\tau| < 1$). Note that the symmetry (A.4) exchanges not only the branch points of these cycles $\sigma_i \to -\sigma_i$, but also exchanges the cycles themselves $\gamma_1 \leftrightarrow \gamma_3$. On the other hand, the symmetry does

---

[24]The reference studies the curve (9) at $Q_b = -1, Q_f = 1$, which are related to $\tau, \kappa$ by $\tau = Q_b/Q_f$ and $\kappa = Q_f^{-1}$. Parameterizing $Q_b = -1 + 2i\rho$ and $Q_f = 1 + i\rho$ and varying $\rho$ from 0 to $+\infty$, one may observe that the left-most branch point in the reference becomes the bottom-right branch point in our picture, the second left-most branch point becomes the top-right one, the top branch point becomes the top-left one and the bottom branch point becomes the bottom-left one. Following the motion of the branch points carefully, one can also check that the BPS saddles $p_i$ in [42, Figure 5] turn into the BPS saddles shown here in Figure 10.

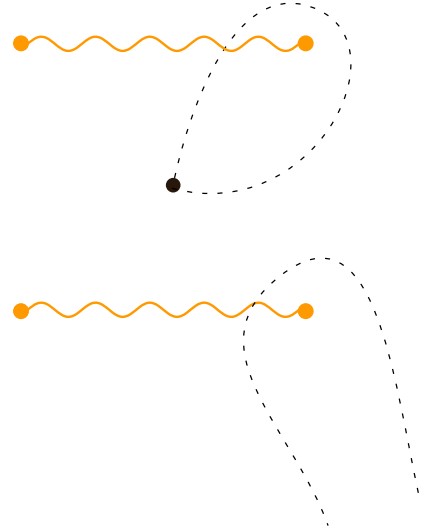

Figure 9: Trivialization of the curve (24). Orange points are square-root branch points, wavy lines are the corresponding branch cuts. Dotted lines are logarithmic branch cuts for the Seiberg-Witten one-form $\lambda$.

not leave the differential invariant by shifts to $\lambda = y\,dx \rightarrow (y + i\pi)\,dx$. However, note that $\oint_{\gamma_1} \lambda = \int_{b_{--}}^{b_{-+}} (\lambda_+ - \lambda_-)$, where $\lambda_\pm = y_\pm(x)\,dx$ are the local values of the Seriberg-Witten differential on the two sheets of $\Sigma$. It follows that the shift of the differential cancels, because the cycle is anti-invariant under the deck involution of $\Sigma$. This implies that $Z_{\gamma_1} = Z_{\gamma_3}$. A similar argument shows that $Z_{\gamma_2} = Z_{\gamma_4}$.

Recalling that $Z_{\gamma_2 + \gamma_4} = \frac{2i}{R} \log \tau$ due to (16)-(17), we have thus shown that

$$Z_{\gamma_1} = Z_{\gamma_3} = \frac{\pi}{R} - \frac{i}{R} \log \tau \,, \qquad Z_{\gamma_2} = Z_{\gamma_4} = \frac{i}{R} \log \tau \,. \tag{A.5}$$

The same result can be obtained if we work with $|\tau| < 1$. These two choices correspond to the chambers described in (29). This analysis shows that the fine-tuned stratum (18) is realized in the physical moduli space of the mirror curve $\Sigma$, precisely in the limit (23).

## B Mirror curve degeneration to half-geometry

In this appendix we describe how to take the limit $\tau \rightarrow 0$ on the curve (9) to obtain (44). We introduce the rescaled coordinate

$$e^{\tilde{x}} = \tau e^x \,, \tag{B.1}$$

and take the limit $\tau \rightarrow 0, x \rightarrow \infty$ keeping $e^{\tilde{x}}$ finite. This gives the curve for the half-geometry

$$(e^y + e^{-y}) + e^{\tilde{x}} - \kappa = 0 \,. \tag{B.2}$$

For later convenience we rewrite curve in new coordinates, by introducing $e^{\tilde{y}} = e^y / e^{y_+}$ with $e^{y_\pm} = \frac{1 \pm \sqrt{1 - 4\kappa^{-2}}}{2\kappa^{-1}} = (e^{y_+})^{\pm 1}$. This gives the curve

$$1 - e^{\tilde{x}} - (\kappa^{-1} e^{y_+}) e^{\tilde{y}} - (\kappa^{-1} e^{-y_+}) e^{-\tilde{y}} = 0 \,. \tag{B.3}$$

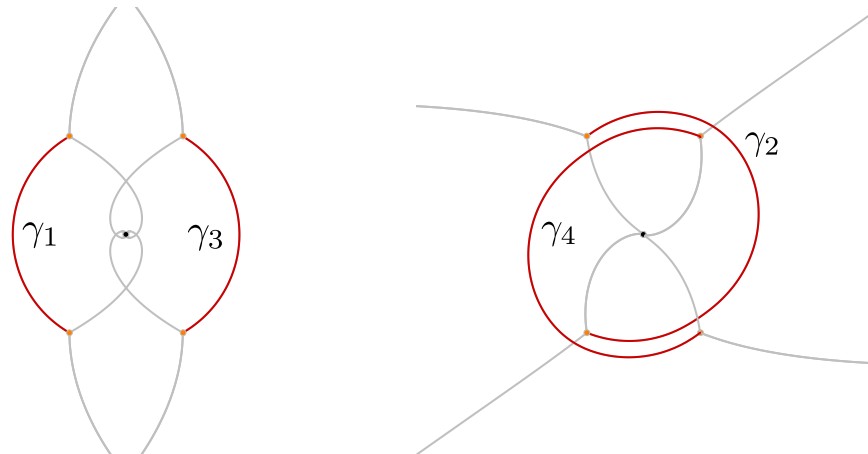

Figure 10: Exponential networks of the curve (24) with $\tau = 2$, at the phase $\arg Z_{\gamma_1}$ (left) and $\arg Z_{\gamma_2}$ (right). Only primary walls are shown, and saddles are highlighted in red. The labels $\gamma_i$ denote the charges (roughly speaking, homology classes) of the cycles obtained by lifting each saddle to $\Sigma$. They correspond to (14) and match with conventions from [42].

Finally we replace the complex modulus $\kappa^{-1}$ by $Q$ via

$$\kappa = \frac{1+Q}{Q^{1/2}} \tag{B.4}$$

and rescale

$$e^{x'} = e^{\tilde{x}}(1+Q), \qquad e^{y'} = e^{\tilde{y}}, \tag{B.5}$$

to obtain a curve described by

$$(1+Q) - e^{x'} - Q\,e^{-y'} - e^{y'} = 0. \tag{B.6}$$

## C WKB periods of first order $\hbar$-difference equations

In the main text we discussed in some detail the computation of compact quantum periods for the half-geometry. Here we collect analogous results for two more geometries: $\mathbb{C}^3$ and the resolved conifold.

$\mathbb{C}^3$

The mirror curve for $\mathbb{C}^3$ can be presented as follows

$$F_{\mathbb{C}^3}(e^x, e^y) = 1 - e^x - e^y = 0, \tag{C.1}$$

in a suitable choice of framing [76, 114]. The quantum curve is obtained in this case by simply replacing $x \to \hat{x}$ and $y \to \hat{y}$, leading to the $\hbar$-difference equation

$$\psi(x+\hbar) = (1-e^x)\psi(x) \quad \Leftrightarrow \quad \mathcal{R}(x;\hbar) = 1 - e^x. \tag{C.2}$$

A solution is then obtained by direct application of (64)

$$\psi(x) = \prod_{n=0}^{\infty} \frac{1}{\mathcal{R}(x+n\hbar;\hbar)} = \prod_{n=0}^{\infty} \frac{1}{1-e^{x+n\hbar}} = \frac{1}{(e^x;e^\hbar)_\infty}, \tag{C.3}$$

where we assumed $\operatorname{Re}\hbar < 0$, and chose the normalization

$$\psi_0(e^x) = 1, \tag{C.4}$$

conventional in open topological string theory [76, 114]. Other normalizatons involving overall multiplication by $\hbar$-periodic factors may also be considered, depending on the desired behavior at $x \to -\infty$. A prescription based on Borel resummation of the asymptotic series expansion of (C.3) was discussed in [28, 65].

We next move on to the computation of quantum periods of the difference equation (C.2). The BPS charge lattice is one-dimensional, and the corresponding generator is the mirror cycle to a D0 brane. Following [41, 50], we can describe this cycle as follows. We observe that (C.1) describes a three-punctured sphere with punctures at $(e^x, e^y) = (0, 1), (1, 0), (\infty, \infty)$. Denoting by $C_z$ a based loop around the puncture at $e^x = z$ oriented counterclockwise, the three based loops around punctures obey $C_\infty \circ C_1 \circ C_0 = 1$, where composition is from the left. Then the BPS cycle is

$$\gamma_{D0} = C_1^{-1} \circ C_0^{-1} \circ C_\infty^{-1} = C_1^{-1} \circ C_0^{-1} \circ C_1 \circ C_0. \tag{C.5}$$

The corresponding quantum period is then obtained by integration of the quantum one-form $\log \mathcal{R}(x; \hbar)$ as in (62). Since the primitive is

$$\frac{1}{\hbar} \int^x \log \mathcal{R}(e^x) \, dy = -\frac{1}{\hbar} \operatorname{Li}_2(e^x), \tag{C.6}$$

the quantum period can be deduced from the monodromy of the dilogarithm function. Recall that $\operatorname{Li}_2$ has the following monodromy properties around paths in the $z$-plane

$$
\begin{aligned}
\operatorname{Li}_2(z) &\to \operatorname{Li}_2(z) - 2\pi i \log z, & &\text{around } C_1, \\
\operatorname{Li}_2(z) &\to \operatorname{Li}_2(z), \qquad \log z \to \log z + 2\pi i, & &\text{around } C_0.
\end{aligned}
\tag{C.7}
$$

The monodromy along $\gamma_{D0}$ then adds up as follows

$$
\begin{aligned}
\operatorname{Li}_2(e^x) &\overset{C_0}{\to} \operatorname{Li}_2(e^x) \\
&\overset{C_1}{\to} \operatorname{Li}_2(e^x) - 2\pi i \log e^x \\
&\overset{C_0^{-1}}{\to} \operatorname{Li}_2(e^x) - 2\pi i (\log e^x - 2\pi i) \\
&\overset{C_1^{-1}}{\to} \operatorname{Li}_2(e^x) + 2\pi i \log e^x - 2\pi i (\log e^x - 2\pi i) \\
&= \operatorname{Li}_2(e^x) - 4\pi^2,
\end{aligned}
\tag{C.8}
$$

so that the quantum period is

$$\Pi_{\gamma_{D0}}(\hbar) = \frac{4\pi^2}{\hbar} = \frac{2\pi R}{\hbar} Z_{\gamma_{D0}}, \tag{C.9}$$

where we refer to the D0 central charge given in (46). In the case of $\mathbb{C}^3$, and more generally in toric geometries without compact four-cycles, the higher order corrections in (59) are absent.

**Resolved conifold**

The mirror curve for the conifold in a suitable choice of framing is

$$F_{\text{conifold}}(e^x, e^y) = 1 - e^y - e^x + Q e^{x+y} = 0. \tag{C.10}$$

This curve is a four-punctured sphere with punctures at $(e^x, e^y) = (0,1),(1,0),(Q^{-1},\infty),$ $(\infty, Q^{-1})$. We work in the phase where $|Q| < 1$, so that the $\mathbb{C}^3$ mirror curve (C.1) is recovered by taking the limit $Q \to 0$ in (C.10). The corresponding $\hbar$-difference equation arising from quantization is

$$(1 - Qe^x)\psi(x + \hbar) - (1 - e^x)\psi(x) = 0 \quad \Leftrightarrow \quad \mathcal{R}(x; \hbar) = \frac{1 - e^x}{1 - Qe^x}. \tag{C.11}$$

Again we can write down a solution directly from (64)

$$\psi(x) = \prod_{n=0}^{\infty} \frac{1}{\mathcal{R}(x + n\hbar; \hbar)} = \prod_{n=0}^{\infty} \frac{1 - Qe^{x+n\hbar}}{1 - e^{x+n\hbar}} = \frac{(Qe^x; e^\hbar)_\infty}{(e^x; e^\hbar)_\infty}, \tag{C.12}$$

where we assumed $\mathrm{Re}\,\hbar < 0$, and chose the normalization (C.4). Other choices of normalizaton, based on Borel resummation of the asymptotic series expansion of (C.12) are discussed in [29].

The BPS charge lattice is now two-dimensional, with generators corresponding to the D2 and $\overline{\text{D2}}$ D0 mirror cycles on $\Sigma$ [50, 62]. Denoting by $C_z$ a counterclockwise loop around the puncture at $e^x = z$, the D2 cycle is

$$\gamma_{D2} = C_0^{-1} \circ C_\infty^{-1} = C_0^{-1} \circ C_{Q^{-1}} \circ C_1 \circ C_0, \tag{C.13}$$

as shown in Figure 11 (also see [62, Figure 2]). The quantum period is obtained by integration of the quantum one-form $\log R(x; \hbar)$ as in (62). Since the primitive is

$$\frac{1}{\hbar} \int^y \log R(e^x)\,dy = \frac{1}{\hbar}\Big[-\mathrm{Li}_2(e^x) + \mathrm{Li}_2(Qe^x)\Big], \tag{C.14}$$

we may obtain the quantum period from the monodromy of the dilogarithm using (C.7)

$$\begin{aligned}
-\mathrm{Li}_2(e^x) + \mathrm{Li}_2(Qe^x) &\xrightarrow{C_0} -\mathrm{Li}_2(e^x) + \mathrm{Li}_2(Qe^x) \\
&\xrightarrow{C_1} -\mathrm{Li}_2(e^x) + 2\pi i\,\log e^x + \mathrm{Li}_2(Qe^x) \\
&\xrightarrow{C_{Q^{-1}}} -\mathrm{Li}_2(e^x) + 2\pi i\,\log e^x + \mathrm{Li}_2(Qe^x) - 2\pi i\,\log(Qe^x) \\
&\xrightarrow{C_0^{-1}} -\mathrm{Li}_2(e^x) + 2\pi i\,(\log e^x + 2\pi i) + \mathrm{Li}_2(Qe^x) - 2\pi i\,(\log(Qe^x) + 2\pi i) \\
&= -\mathrm{Li}_2(e^x) + \mathrm{Li}_2(Qe^x) - 2\pi i\,\log Q.
\end{aligned} \tag{C.15}$$

The D2 quantum period is then

$$\Pi_{\gamma_{D2}} = -\frac{2\pi i}{\hbar}\log Q = \frac{2\pi R}{\hbar} Z_{\gamma_{D2}}. \tag{C.16}$$

To complete the basis of quantum periods we need a second, linearly independent, cycle which we take to be $\gamma_{D0}$. The D0 cycle may be obtained by noting that (C.10) is a pair of trinions glued along a tube, and by recalling that each trinion is a copy of the mirror curve of $\mathbb{C}^3$. In the phase we are studying, characterized by $|Q| < 1$ it is natural to decompose (C.10) into two trinions glued by a long thin neck separating $x = 0,1$ from $x = Q^{-1}, \infty$. We then embed the D0 cycle (C.5) into, say, the left trinion

$$\gamma_{D0} = C_1^{-1} \circ C_0^{-1} \circ C_{\text{neck}}^{-1} = C_1^{-1} \circ C_0^{-1} \circ C_1 \circ C_0. \tag{C.17}$$

To really justify our identification of the D0 charge, one should prove that there is a calibrated cycle in this homology class. This is indeed the case, as shown in [62, Figure 2]. The computation of the D0 quantum period proceeds in a similar way as for $\mathbb{C}^3$, by studying monodromies of (C.14). The result is of course again (C.9).

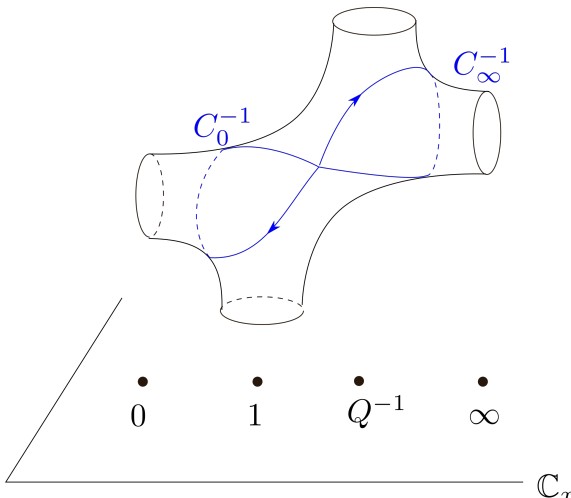

Figure 11: The cycle $\gamma_{D2}$ on the mirror curve of the conifold, shown as a covering over the $x$ plane. Labels of punctures denote the values of $e^x = 0, 1, Q^{-1}, \infty$ respectively.

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
