# Peer review of "The threefold way to quantum periods: WKB, TBA equations and q-Painleve"

_SciPost Physics, doi:SciPost Phys. 15, 112 (2023)_

## Round 1 · Referee Report · Anonymous (Referee 1) · 2022-12-31

Report

Review of "The threefold way to quantum periods: WKB, TBA equations and q-Painleve" by Fabrizio Del Monte and Pietro Longhi:

The WKB method, the TBA equations and the Painleve equations are three a-priory different methods to study quantum curves. Although their relation is well-studied for four-dimensional N=2 theories of class S, there are only limited results for five-dimensional N=1 theories.

This is another exciting paper by Fabrizio and Pietro in which they study the relations between these three methods for the 5d $\mathcal{N}=1$ $SU(2)$ Yang-Mills theory on $S^1 \times \mathbb{R}^4$, corresponding to the local $\mathbb{P}^1 \times \mathbb{P}^1$ geometry. The results of this paper are obtained by studying a particular chamber (called the collimination chamber) of the moduli space of stability conditions for the 5d $SU(2)$ theory, in which the BPS spectrum is rather simple, and summarized by the stability condition (2.27) and Figure 4. Important in the paper is the (affine) $\mathbb{Z}$-action $T$ acting on the central charges of the BPS states as in eqn (2.24).

A small stratum of this chamber, which is referred to as the fine-tuned stratum of the collimination chamber, may be geometrically realized on the physical moduli space of the 5d theory. In this stratum the central charges of the four generating BPS states are pairwise identified, the TBA equations (in the conformal limit) can be solved exactly, and it turns out that none of the quantum periods receive $\epsilon$-corrections. This leads to an equality between the WKB quantum periods and the TBA quantum periods, which is argued to extend across the whole physical moduli space in the paragraph below eqn (4.1).

In the larger collimation chamber the quantum periods do receive $\epsilon$-corrections, and the WKB quantum periods are difficult to compute. Yet, using the affine symmetry $T$ it is possible to bring the TBA equations in the form of a $q$-Painleve equation. Since the general solution to the $q$-Painleve equation is known to be a ratio of so-called dual instanton partition functions for the 5d $SU(2)$ theory, it is only needed to identify the correct values for the integration constants to find the exact solutions to the TBA equations in the collimination chamber. This leads to the conclusion that the TBA equations in the collimination chamber are equal to the $q$-Painleve solutions evaluated at $y = q^{1/4}$. In section 5.4 it is conjectured that this relation may be extended across the whole moduli space.

The paper gives a very interesting new perspective on the relation between WKB/TBA quantum periods and the Painleve integrable system, and is clearly written. It therefore deserves to be published in SciPost, up to a few minor corrections.

Some questions/remarks:

$*$ As remarked later in the paper, the relation between the WKB/TBA quantum periods and the q-Painleve solutions relates an object that transforms non-trivially when the argument of $h$ is changed to an object that is defined non-perturbatively in $q$. I think that this deserves some attention in the introduction as well. As mentioned in the conclusion, an explanation could be that in the argument to find the correct values for the integration constants for the $q$-Painleve solutions, in section 5.3, one implicitly focusses on a special phase of $h$ corresponding to a Fenchel-Nielsen type network (similar to [40] and https://arxiv.org/abs/2109.14699).

$*$ In eqn (3.13) the function $S(x)$ needs an extra $dx$.

$*$ On p13 you mention Borel summability in the context of difference equations. What do we currently know about this?

$*$ At the end of section 3.3 it is suggested that all solutions to the difference equation may be found using Borel summation. Is that indeed true?

$*$ At the end of section 3.3 it is mentioned that normalization factors do not affect the definition of the quantum periods. The argument here is probably the clever (yet, as it is written down, $h$-perturbative) reasoning in section 3.2, which computes the quantum periods purely in terms of the eigenvalue $R$. Because of the structure of the paragraph, this seems to suggest that quantum periods do not depend on the angular sector of the Borel plane, which is clearly not true, unless they do not receive quantum corrections. This will be important in establishing the precise relation to $q$-Painleve later in the paper.

$*$ A similar remark also applies to the sentence "quantum periods .. polarization" above equation (3.25). I think this relation might be affected by non-perturbative corrections in $h$. Of course this is not relevant when the quantum periods considered are not quantum corrected.

$*$ At the end of section 3.5, "differently" -> "Differently". Yet, is there really a complication in the differential case to pass from Ricatti solutions for second order ODE's to higher order ones? (see for instance https://arxiv.org/abs/1906.04271)

$*$ In section 4.1, "connectng" -> "connecting".

$*$ Perhaps it is useful for the reader to also cite works in section 4.4 with (numerical) TBA solutions for coupled BPS structures.

$*$ End of section 4 "disctete" -> "discrete".

$*$ Under eqn (5.33) "constrast" -> "contrast".

$*$ Section 5.4 could be made stronger. Especially observation 3 requires some extra explanations and does not really belong to this list of "observations". But also the argument in the second paragraph of this section is not very clear to me. Do you want to argue that the exact TBA solutions will become much more complicated when leaving the fine-tuned stratum because the spectrum of BPS states becomes wild, and that the solutions to the Painleve equations similarly drastically change behaviour, so that we can still believe that there is an equality between the two?

$*$ Section 5.4: "plausiblle" and "neighbouhood".

$*$ In section 6 just before the section "Beyond local $\mathbb{P}^1 \times \mathbb{P}^1$" and right after the reference [68], it might be useful to add the references [69] and https://arxiv.org/abs/2109.14699. These papers interpret the non-perturbative $h$-corrections to the 4d WKB quantum periods in terms of 4d BPS states.

  • validity: -
  • significance: -
  • originality: -
  • clarity: -
  • formatting: -
  • grammar: -

Author:  Pietro Longhi  on 2023-03-30  [id 3521]

(in reply to Report 1 on 2022-12-31)

We thank the referee for their careful reading of the manuscript and for the suggestions for improvement.
In regards to their questions, we addressed them as follows:

∗ As remarked later in the paper, the relation between the WKB/TBA quantum periods and the q-Painleve solutions relates an object that transforms non-trivially when the argument of h is changed to an object that is defined non-perturbatively in q. I think that this deserves some attention in the introduction as well. As mentioned in the conclusion, an explanation could be that in the argument to find the correct values for the integration constants for the q-Painleve solutions, in section 5.3, one implicitly focusses on a special phase of h corresponding to a Fenchel-Nielsen type network (similar to [40] and https://arxiv.org/abs/2109.14699).

We added citations to [Hollands-Ruter-Szabo 2109.14699] and [Grassi, Hao,Neitzke 2105.03777] on page 39.

We added a remark to the introduction below eq. (1.7), and also a longer paragraph later in the main text (Rmk 3 on page 37).

∗On p13 you mention Borel summability in the context of difference equations. What do we currently know about this?

We are not aware of any treatment of this subject for higher-order \hbar-difference equations. We added some pointers to literature relevant to the first order case, in footnote 8.

∗ At the end of section 3.3 it is suggested that all solutions to the difference equation may be found using Borel summation. Is that indeed true?

At the end of section 3.3 we explain how to use Borel summation to obtain solutions to the difference equation with various types of asymptotics, and while keeping into account possible ambiguities due to \hbar-periodic factors. We did not delve into the question of whether this class of solutions is fully exhaustive.

∗ At the end of section 3.3 it is mentioned that normalization factors do not affect the definition of the quantum periods. The argument here is probably the clever (yet, as it is written down, h-perturbative) reasoning in section 3.2, which computes the quantum periods purely in terms of the eigenvalue R. Because of the structure of the paragraph, this seems to suggest that quantum periods do not depend on the angular sector of the Borel plane, which is clearly not true, unless they do not receive quantum corrections. This will be important in establishing the precise relation to
q-Painleve later in the paper.

We added a remark to clarify this potential confusion, indeed quantum periods are sensitive to the phase of \hbar through Stokes automorphisms.

∗ A similar remark also applies to the sentence "quantum periods .. polarization" above equation (3.25). I think this relation might be affected by non-perturbative corrections in h. Of course this is not relevant when the quantum periods considered are not quantum corrected.

There might be such subtleties for coupled systems when computing quantum periods in different choices of polarisations. But in the case of (3.25) we are working with an uncoupled BPS structure, the one pertinent to the "half-geometry". For this reason there should be no such ambiguities.

∗ At the end of section 3.5, "differently" -> "Differently". Yet, is there really a complication in the differential case to pass from Ricatti solutions for second order ODE's to higher order ones? (see for instance https://arxiv.org/abs/1906.04271).

Indeed, it is plausible that a general formula similar (in spirit) to (3.40) can be written down also for the differential case. We adjusted the remark accordingly.

∗ Perhaps it is useful for the reader to also cite works in section 4.4 with (numerical) TBA solutions for coupled BPS structures.

It is unclear to us whether it would be really beneficial to add such references at this point in the draft, since Section 4.4 presents an exact analytic solution.
Instead, we cite numerical work for TBA equations associated to 4d theories elsewhere, e.g. Grassi-Gu-Marino [67]. We are unfortunately not aware of literature dealing with numerical solutions to TBA equations for local P1xP1.

∗ Section 5.4 could be made stronger. Especially observation 3 requires some extra explanations and does not really belong to this list of "observations". But also the argument in the second paragraph of this section is not very clear to me. Do you want to argue that the exact TBA solutions will become much more complicated when leaving the fine-tuned stratum because the spectrum of BPS states becomes wild, and that the solutions to the Painleve equations similarly drastically change behaviour, so that we can still believe that there is an equality between the two?

We believe that Painlev\'e equations are strictly related to the non-wild spectrum that is found in the collimation chamber (also beyond the fine-tuned stratum), the connection being worked out in section 4.5.
In the latest version we have removed previous speculations about extensions of the q-Painlev'e solutions to wild regions, and made more precise the discussion about the collimation chamber.
In fact, we made significant changes in the new version of the manuscript concerning deformations away from the fine-tuned stratum. Among these, section 5.4 has been dropped.
Starting with (5.24) of the latest version, the definition of integration constants has been rectified from the previous (erroneous) one. See in particular eq.s (5.30)-(5.33) in the new version.
Also note section 2.2.3 where two types of deformations of the fine-tuned stratum (within the collimation chamber) have now been introduced.

∗In section 6 just before the section "Beyond local P1×P1" and right after the reference [68], it might be useful to add the references [69] and https://arxiv.org/abs/2109.14699. These papers interpret the non-perturbative h-corrections to the 4d WKB quantum periods in terms of 4d BPS states.

Thank you for the suggestions, we have implemented these.

---

## Round 1 · Referee Report · Anonymous (Referee 2) · 2023-2-27

Report

Dear Authors, dear Editor, my report is attached.

Attachment

  • validity: -
  • significance: -
  • originality: -
  • clarity: -
  • formatting: -
  • grammar: -

Author:  Pietro Longhi  on 2023-03-30  [id 3520]

(in reply to Report 2 on 2023-02-27)

We thank the referee for their careful reading of the manuscript and for the suggestions for improvement. In regards to their questions, we addressed them as follows:

  1. We corrected this, thanks for the suggestion.
  2. We thank the referee for pointing out this elegant argument, and we have implemented this in appendix A.
  3. Indeed there is an ambiguity in the definition of the tau function / dual partition function. However, to our understanding this ambiguity does not affect the X-cluster variables. We have added a footnote to clarify this around eq. (5.21).
  4. The spectral determinant is defined on the `physical' locus, while the TBA equations can be formally defined in the larger space of stability conditions. It is true that \rho, \delta, and the spectral determinant are all related to 1-parameter solutions, but their relation is unclear because we do not know if \rho and ]delta deformations belong to the physical slice (see comment WKB vs TBA on page 37).

---

## Round 2 · Referee Report · Anonymous (Referee 3) · 2023-6-3

Report

I'd like the thank the authors for the careful revision, and am happy to recommend the paper for publishing.

---

## Round 2 · Referee Report · Anonymous (Referee 4) · 2023-7-5

Report

Dear Authors, dear Editor,

I apologize for the long delay and thank the authors for the improvements done in the paper. I think that now it can be published as is.

---

## Round 2 · Author Response

We have addressed the comments and questions raised by the referees.

---

## Round 2 · List of Changes

See replies to each of the referees for detailed comments on changes.

---

## Editorial Decision

published